# Elastic Attention: Test-time Adaptive Sparsity Ratios for Efficient Transformers

Zecheng Tang [* 1 2]   Quantong Qiu [* 1 2]   Yi Yang [1 2]   Zhiyi Hong [1 2]   Haiya Xiang [1 2]   Kebin Liu [3]   Qingqing Dang [3]   Juntao Li [1 2]   Min Zhang [1]

## Abstract

The quadratic complexity of standard attention mechanisms poses a significant scalability bottleneck for large language models (LLMs) in long-context scenarios. While hybrid attention strategies that combine sparse and full attention within a single model offer a viable solution, they typically employ static computation ratios (i.e., fixed proportions of sparse versus full attention) and fail to adapt to the varying sparsity sensitivities of downstream tasks during inference. To address this issue, we propose *Elastic Attention*, which allows the model to dynamically adjust its overall sparsity based on the input. This is achieved by integrating a lightweight *Attention Router* into the existing pretrained model, which dynamically assigns each attention head to different computation modes. Within only 12 hours of training on $8 \times$A800 GPUs, our method enables models to achieve both strong performance and efficient inference (see Figure 1[1]). Experiments across three long-context benchmarks on widely-used LLMs demonstrate the superiority of our method.

## 1. Introduction

Large language models (LLMs) have demonstrated remarkable capabilities in processing long-context sequences (Liu et al., 2025a;b; Mei et al., 2025). However, the quadratic computational and memory complexity of standard full attention (**FA**) mechanisms (Vaswani et al., 2017) poses a significant scalability bottleneck as the context window continues to expand. Sparse attention (**SA**) mechanisms (Child, 2019; Zaheer et al., 2020) represent an effective strategy for mitigating this limitation by selectively attending to a subset

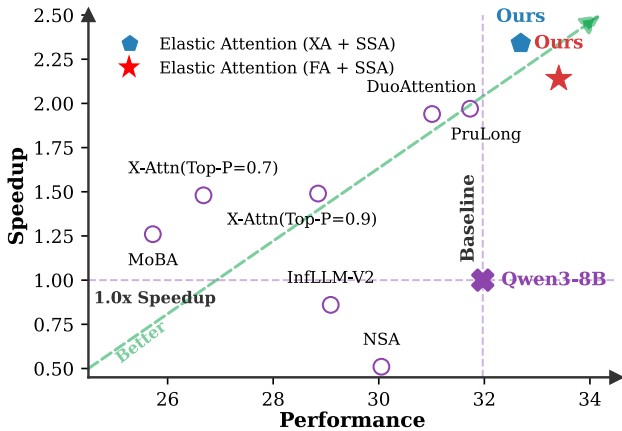

*Figure 1.* Comparison between Elastic Attention (ours) and existing approaches on LongBench-V2 (Bai et al., 2025). "(XA+SSA)" and "(FA+SSA)" denote our different settings.

of critical tokens, thereby substantially reducing computational overhead and improving inference throughput.

To balance the trade-off between reduced computational cost and preserved model quality, i.e., achieving performance comparable to FA, existing methods typically adopt hybrid modeling strategies that integrate both SA and FA within a single model (Zhang et al., 2025a; Xiao et al., 2025). Yet, downstream task performance is susceptible to the proportion between those two attention computation modes (see our preliminary study in § 2.2), and identifying an appropriate proportion typically requires extensive task-specific validation or tuning, which severely limits robustness and general applicability (Peng et al., 2025a).

In this work, we show that downstream tasks naturally fall into two categories: (1) sparsity-robust tasks (e.g., summarization), whose performance remains stable across a wide range of sparsity levels, and (2) sparsity-sensitive tasks (e.g., question answering), which suffer substantial degradation beyond a certain sparsity threshold. Such an observation suggests that a model should only differentiate between these two task regimes and allocate the appropriate sparsity, instead of painstakingly learning distinct sparsity configurations for each task. Going beyond this, we propose *Elastic Attention*, which enables the model to automatically adjust its overall sparsity during the prefill stage to accommodate

[1]Soochow University, China [2]LCM Laboratory [3]Baidu Inc, China. Correspondence to: Juntao Li <ljt@suda.edu.cn>.

*Proceedings of the 43rd International Conference on Machine Learning*, Seoul, South Korea. PMLR 306, 2026. Copyright 2026 by the author(s).

[1]NSA (Yuan et al., 2025) and InfLLM-V2 (Zhao et al., 2025) impose architectural constraints, incompatible with Qwen3-8B

the two aforementioned task categories. The core of our approach is a lightweight **Attention Router** module integrated into existing transformer architectures.

Specifically, the Attention Router operates analogously to Mixture-of-Experts (Shazeer et al., 2017): it uses the input hidden states to perform head-wise routing, dynamically assigning each attention head to either FA or SA computation mode, thereby enabling dynamic adjustment of the model's overall sparsity. Notably, the Attention Router introduces negligible overhead, adding only 0.27M parameters per layer (assuming a head dimension of 128), which preserves both inference efficiency and computational cost. To optimize the Attention Router, we adopt a continuous relaxation scheme based on the Gumbel-Softmax (Jang et al., 2016) strategy, which alleviates the training–inference discrepancy arising from discrete routing decisions. Furthermore, to stabilize optimization, we employ a straight-through estimator (STE) (Bengio et al., 2013) trick for gradient propagation. During inference, different attention heads within the same layer may be routed to different computation modes. We design a fused kernel that enables all routed heads to be processed simultaneously in a single forward pass.

We evaluate our method on 3 widely-used and cutting-edge LLMs (Qwen3-series (Yang et al., 2025) and Llama-3.1-8B-Instruct (Grattafiori et al., 2024) models), considering both streaming (Xiao et al., 2024b) and block-sparse (Guo et al., 2024) SA patterns. Across a diverse set of long-context benchmarks (real-world, synthetic retrieval, and long-form reasoning tasks), our method not only learns to allocate different sparsity levels to different tasks but also yields superior performance compared to baselines. Notably, this gain is achieved under limited training budgets, requiring only 12 hours of training on 8×A800 GPUs without modifying the backbone model's parameters. Furthermore, we conduct an in-depth analysis of the Attention Router, studying the impact of its architectural design and parameter capacity, as well as providing detailed training monitoring metrics[2].

## 2. Preliminary

### 2.1. Retrieval Heads and Sparse Heads

**Retrieval Heads and FA**  Retrieval heads are a class of attention heads specialized in capturing contextually relevant tokens from long sequences, enabling modern LLMs to operate effectively in long-context processing tasks (Wu et al., 2024). In practice, retrieval heads typically adopt the standard FA computation mode. Given the Query ($Q$), Key ($K$), and Value ($V$) states, the computation of FA is :

$$\mathcal{O}_r = \text{Softmax}\left(QK^\top\right)V, \tag{1}$$

---

[2]Our code is available at: https://github.com/LCM-Lab/Elastic-Attention. Besides, Elastic Attention can also be implemented by the PaddlePaddle framework.

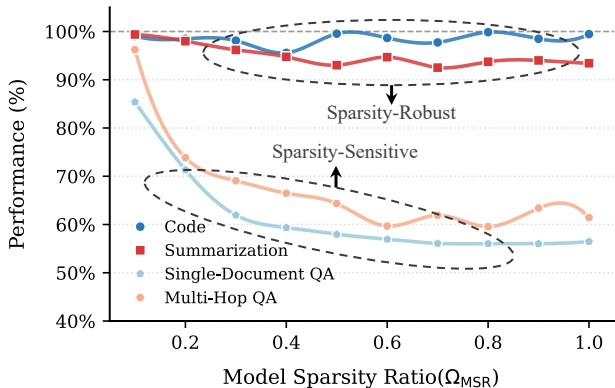

*Figure 2.* Trend of model performance as the hybrid model sparsity ratio ($\mathbf{\Omega_{MSR}}$) increases. We report model performance as a relative percentage score with respect to that of FA.

where we omit the scaling factor for simplicity.

**Sparse Heads and SA**  Compared to the retrieval heads, sparse heads reduce computational cost by leveraging SA mode, which retains only a fraction of the most relevant $K$ and $V$ (e.g., 20% of key and value states). The computation of SA can be defined as:

$$\mathcal{O}_s = \text{Softmax}\left(Q\tilde{K}^\top\right)\tilde{V}, \tag{2}$$

where $\tilde{K}$ and $\tilde{V}$ are partial elements in $K$ and $V$.

**Hybrid Head Mechanism**  While SA substantially reduces computation, it may lead to performance degradation (Lu et al., 2025b). To balance efficiency and performance, recent methods adopt a *hybrid heads* design that mixes retrieval and sparse heads within the same model (Xiao et al., 2025; Yu et al., 2025; Bhaskar et al., 2025). Specifically, for a Transformer with L layers and H key–value heads per layer[3], the $h$-th head in the $\ell$-th layer is assigned a computation type: $\pi^{(\ell,h)} \in \{\text{FA}, \text{SA}\}$, and the final output per layer ($\mathbf{O}^{(\ell)}$) can be written as:

$$\begin{cases} \mathbf{O}^{(\ell)} = \text{Concat}\left(\mathcal{O}^{(\ell,1)}, \mathcal{O}^{(\ell,2)}, \ldots, \mathcal{O}^{(\ell,H)}\right), \\ \mathcal{O}^{(\ell,h)} = \begin{cases} \mathcal{O}_r, & \pi^{(\ell,h)} = \text{FA}, \\ \mathcal{O}_s, & \pi^{(\ell,h)} = \text{SA}, \end{cases} \end{cases} \tag{3}$$

**Sparsity Ratio Computation**  Under the hybrid-head mechanism, given a model $f_\theta$, we define two types of model sparsity ratios from two perspectives: (1) the proportion of sparse attention heads, and (2) the proportion of tokens attended to by those heads.

---

[3]Here we adopt grouped query attention (GQA) (Ainslie et al., 2023) definition instead of multi-head attention (MHA), as GQA is widely used in modern LLM architectures.

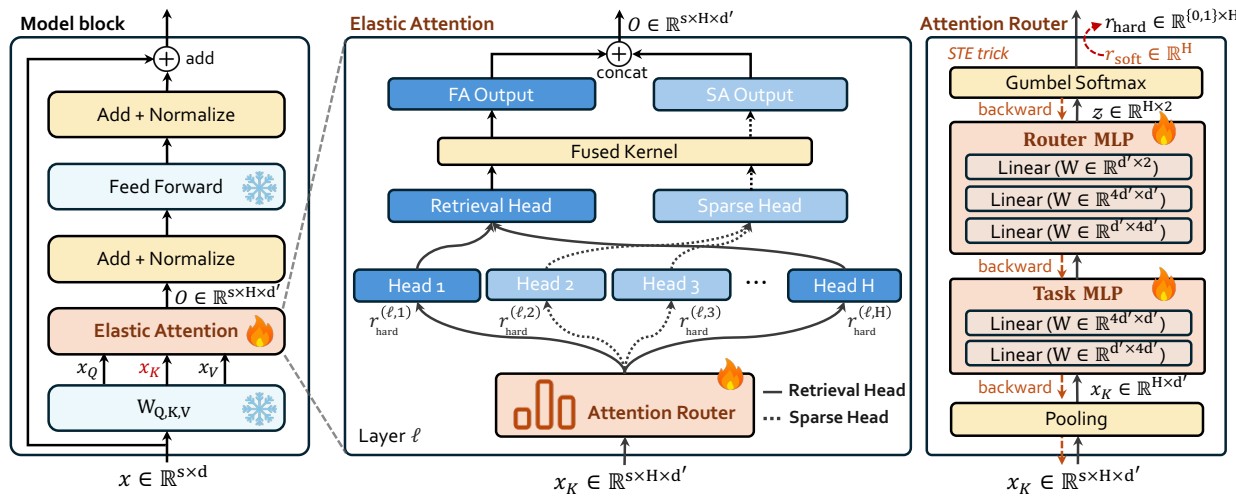

(a) Architecture of model block  (b) Information flow of Elastic Attention and Attention Router  (c) Design of Attention Router

*Figure 3.* Illustration of our proposed Elastic Attention. (a) shows the adapted model block with frozen backbone parameters; (b) details the dynamic assignment of heads via the Attention Router module; (c) presents the lightweight design of the Attention Router.

**Definition 2.1** (Model Sparsity Ratio, $\Omega_{\text{MSR}}$). $\Omega_{\text{MSR}}$ measures the fraction of sparse heads in the model:

$$\Omega_{\text{MSR}}(f_\theta) = \frac{1}{H} \frac{1}{L} \sum_{h=1}^{H} \sum_{\ell=1}^{L} \mathbb{I}\left[\pi^{(\ell,h)} = \text{SA}\right], \quad (4)$$

where $\mathbb{I}[\cdot]$ is the indicator function.

**Definition 2.2** (Effective Sparsity Ratio, $\Omega_{\text{ESR}}$). $\Omega_{\text{ESR}}$ measures the effective sparsity of the entire model, taking both the sparse heads and their corresponding sparsity patterns into account. It can be calculated as:

$$\Omega_{\text{ESR}}(f_\theta) = \frac{1}{H} \frac{1}{L} \sum_{h=1}^{H} \sum_{\ell=1}^{L} \rho^{(\ell,h)}, \quad (5)$$

where $\rho^{(\ell,h)} \in [0, 1)$ denotes the pruning ratio of each head, with $\rho^{(\ell,h)} = 0$ for heads with FA, since no tokens are pruned, and $\rho^{(\ell,h)} = \rho_{\text{SA}}$ for heads with SA[4].

### 2.2. Impact of $\Omega_{\text{MSR}}$ on Downstream Tasks

While existing hybrid head mechanisms have achieved strong performance–efficiency trade-offs, they suffer from a critical limitation: *the $\Omega_{MSR}$ is fixed at inference time, and its relationship with downstream task performance remains unexplored.* In this section, we investigate the impact of varying $\Omega_{\text{MSR}}$ on different downstream tasks.

**Settings** We experiment with the Llama3.1-8B-Instruct model (Grattafiori et al., 2024) and follow Wu et al. (2024) to identify retrieval heads in the model and rank them by

their activation frequency. Based on this ranking, we progressively replace retrieval heads with sparse heads, thereby increasing the $\Omega_{\text{MSR}}$. We evaluate the above resulting model on LongBench (Bai et al., 2024), which consists of 6 distinct long-context downstream tasks. For different values of $\Omega_{\text{MSR}}$, we report the model performance as a percentage relative to that of the backbone model (full FA). For instance, at an $\Omega_{\text{MSR}} = 20\%$, the resulting model may achieve 73.85% of the backbone model's performance ($\Omega_{\text{MSR}} = 0\%$). We provide background of retrieval head identification and implementation details in Appendix B.

**Results** As shown in Figure 2, the tasks can be broadly categorized into two categories. The first category, which we refer to as ***sparsity-robust tasks***, exhibits performance that is largely insensitive to changes in $\Omega_{\text{MSR}}$. This is because these tasks can be solved using only coarse-grained contextual information, e.g., summarization. The second category, ***sparsity-sensitive tasks***, shows a sharp decline in performance once $\Omega_{\text{MSR}}$ exceeds a certain threshold. Such tasks require retrieving fine-grained evidence from the context to produce accurate outputs, as is typical in question-answering tasks. Therefore, *regardless of the number of task types, they can all be mapped into one of these two categories*. Under this setting, the model only needs to determine whether a task requires fine-grained information and then apply the corresponding attention computation mode accordingly.

## 3. Elastic Attention

We provide an overview of our approach in Figure 3. The sub-figure 3(a) illustrates the overall architecture integrating our Elastic Attention module, whose core component is an Attention Router mechanism. Notably, except for the Elastic

---

[4]Different SA strategies correspond to specific $\rho_{\text{SA}}$, e.g., if SA only attends to 10% tokens along the sequence dimension, then $\rho_{\text{SA}} = 0.9$, meaning that 90% of tokens are pruned.

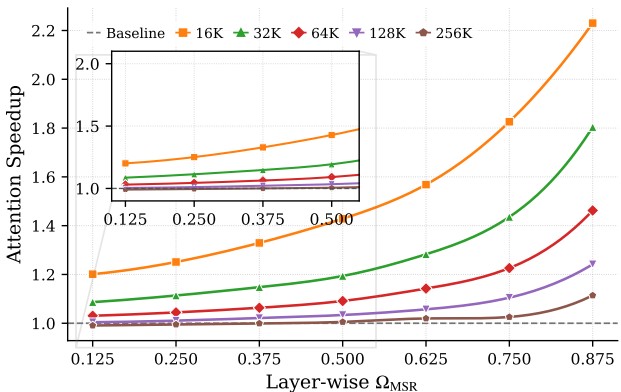

*Figure 4.* Comparison of our fused kernel with a Torch-based sequential implementation for layer-wise hybrid attention.

Attention components, all backbone model parameters are frozen during training.

### 3.1. Introduce Attention Router

**Information Flow of Attention Router**    As illustrated in sub-Figure 3(b), the attention router operates in a manner analogous to a Mixture-of-Experts (MoE) (Shazeer et al., 2017) gating mechanism: it determines, based on the input hidden states, which attention computation mode (i.e., FA or SA) is assigned to each key–value head. Specifically, the Key hidden states ($\boldsymbol{x}_K \in \mathbb{R}^{s \times H \times d'}$, where $s$ denotes sequence length, $d'$ denotes head dimension) are fed into the attention router. Then, it produces a hard binary decision ($r_{\text{hard}}^{(\ell,h)} \in \{0,1\}$) for each head, indicating whether this head employs FA ($r_{\text{hard}}^{(\ell,h)} = 0$) or SA ($r_{\text{hard}}^{(\ell,h)} = 1$) computation mode. Each head then adopts its allocated attention computation mode, and the resulting head outputs are concatenated along the feature dimension to form the final output.

**Architecture Design of Attention Router**    We show the architecture of the attention router in sub-Figure 3(c), which consists of two components: a **task MLP** and a **router MLP**. The task MLP is designed to infer task-specific characteristics from the input hidden states, while the router MLP leverages these task-aware representations to determine the attention computation mode for each head. More concretely, given the Key hidden states ($\boldsymbol{x}_K$), the attention router first applies pooling along the sequence dimension to obtain a task representation $\boldsymbol{x}'_K \in \mathbb{R}^{H \times d'}$.[5] The pooled latent $\boldsymbol{x}'_K$ is then fed into the task MLP and the router MLP to produce a head-wise routing logit matrix ($z \in \mathbb{R}^{H \times 2}$), which is then converted into a binary decision $r_{\text{hard}}^{(\ell,h)}$ for each head via a softmax-based classifier.

[5]In practice, we avoid pooling over the full sequence, as long context introduces substantial redundancy that degrades the quality of task representation. We investigate this in Appendix G.5.

### 3.2. Optimization-based Head Identification

**Attention Router Optimization**    Notably, to ensure consistency between training and inference, where inference uses hard routing decisions (i.e., binary classification), we also adopt hard routing during training. This introduces two challenges: (i) *how to make the softmax-based routing distribution closely approximate hard routing decisions*, and (ii) *how to handle the non-differentiability introduced by hard routing decisions*. To address the first issue, we adopt a continuous relaxation scheme based on the Gumbel–Softmax (Jang et al., 2016) with annealing, which gradually sharpens the routing distribution during training toward discrete assignments. Specifically, we feed $z$ to the Gumbel–Softmax and obtain a soft routing matrix $r_{\text{soft}}^{(\ell)} \in \mathbb{R}^{H \times 2}$, where each column corresponds to the routing probabilities of all heads in layer $\ell$. Hard routing decisions are then obtained by applying $\arg\max$ over the second dimension:

$$r_{\text{hard}}^{(\ell,h)} = \underset{c \in \{0,1\}}{\arg\max} \; r_{\text{soft}}^{(h,c)}, \qquad \forall h \in \{1, \dots, H\}. \quad (6)$$

For the second issue, since $\arg\max$ operation is non-differentiable and would block gradient flow, we employ a straight-through estimator (STE) strategy (Bengio et al., 2013), where $r_{\text{hard}}^{(\ell,h)}$ can be rewritten as:

$$r_{\text{hard}}^{(\ell,h)} = r_{\text{hard}}^{(\ell,h)} + \left[ r_{\text{soft}}^{(\ell,h)} - \text{gradient\_detach}\left( r_{\text{soft}}^{(\ell,h)} \right) \right], \quad (7)$$

which enables gradients to propagate through the soft routing distribution in the backward process while preserving hard routing behavior in the forward pass. We provide more details and theoretical explanations in Appendix E.

**Training Objective**    We use the $\Omega_{\text{MSR}}$ metric to calculate the predicted sparsity during training and compare it against the task-specific target sparsity $\boldsymbol{t}$. Notably, instead of enforcing a fixed $\boldsymbol{t}$ for each task, we impose *task-dependent non-tight constraints* with predefined lower and upper bounds since the optimal sparsity for a given task is unknown. To ensure that introducing sparsity does not degrade the language modeling capability, we build a min-max optimization term upon the standard language modeling objective with a sparsity regularization term. Given an input $\mathcal{X}$ and corresponding label $\mathcal{Y}$, the training objective is:

$$\max_{\lambda_1, \lambda_2} \min \underbrace{L_{\text{language}}(\mathcal{X})}_{\text{language modeling}} + \underbrace{\lambda_1 L_{\text{diff}}(\mathcal{X}) + \lambda_2 L_{\text{diff}}^2(\mathcal{X})}_{\text{sparsity regularization}},$$

$$\Rightarrow \begin{cases} L_{\text{language}} = \text{CE\_Loss}\left( \mathcal{Y} \mid f_\theta(\mathcal{X}) \right), \\ L_{\text{diff}} = \Omega_{\text{MSR}}\left( f_\theta(\mathcal{X}) \right) - \boldsymbol{t}, \end{cases}$$

$$(8)$$

where $\text{CE\_Loss}$ denotes the standard cross-entropy loss for language modeling, and $\lambda_1, \lambda_2$ are task-specific trainable Lagrange multipliers optimized via gradient ascent (Bhaskar

*Table 1.* Performance on LongBench-E (Bai et al., 2024). We report average performance (Perf.) and $\Omega_{\mathrm{MSR}}$ per task category. The 1st and the 2nd performance in each comparison group are highlighted with **bold font** and underlined, respectively.

| Method | S-Doc QA | | M-Doc QA | | Summ | | In-Context | | Synthetic | | Code | | Avg. | |
|---|---|---|---|---|---|---|---|---|---|---|---|---|---|---|
| | Perf. | $\Omega_{\mathrm{MSR}}$ | Perf. | $\Omega_{\mathrm{MSR}}$ | Perf. | $\Omega_{\mathrm{MSR}}$ | Perf. | $\Omega_{\mathrm{MSR}}$ | Perf. | $\Omega_{\mathrm{MSR}}$ | Perf. | $\Omega_{\mathrm{MSR}}$ | Perf. | $\Omega_{\mathrm{MSR}}$ |
| Qwen3-4B backbone model | | | | | | | | | | | | | | |
| Qwen3-4B | 43.69 | - | 38.48 | - | 28.46 | - | 66.21 | - | 49.59 | - | 54.38 | - | 48.45 | - |
| + InfLLM-V2 | 43.30 | - | 34.97 | - | **29.95** | - | 64.19 | - | 38.54 | - | **59.58** | - | 46.68 | - |
| + DuoAttention | 41.73 | 0.70 | 35.72 | 0.70 | 28.47 | 0.70 | 64.59 | 0.70 | 47.17 | 0.70 | 53.91 | 0.70 | 46.95 | 0.70 |
| + PruLong | 41.58 | 0.70 | 37.58 | 0.70 | 28.53 | 0.70 | 64.80 | 0.70 | 47.23 | 0.70 | 53.20 | 0.70 | 47.19 | 0.70 |
| + Elastic Attention (FA-SSA) | 42.20 | 0.66 | 38.86 | 0.68 | 28.50 | 0.76 | **65.73** | 0.73 | **48.43** | 0.71 | 54.34 | 0.82 | **48.08** | 0.73 |
| + Elastic Attention (FA-XA) | **44.40** | 0.68 | **39.42** | 0.71 | 28.49 | 0.82 | 65.26 | 0.76 | 44.35 | 0.74 | 54.29 | 0.87 | 47.59 | 0.76 |
| Qwen3-8B backbone model | | | | | | | | | | | | | | |
| Qwen3-8B | 45.57 | - | 51.59 | - | 28.34 | - | 66.64 | - | 50.16 | - | 61.20 | - | 52.16 | - |
| + InfLLM-V2 | 42.20 | - | 42.33 | - | **29.58** | - | 64.55 | - | 45.87 | - | 59.58 | - | 49.03 | - |
| + DuoAttention | 45.45 | 0.70 | 44.52 | 0.70 | 28.16 | 0.70 | 66.42 | 0.70 | 47.50 | 0.70 | 62.38 | 0.70 | 50.67 | 0.70 |
| + PruLong | 46.05 | 0.70 | 46.88 | 0.70 | 28.30 | 0.70 | 66.61 | 0.70 | 48.66 | 0.70 | 62.24 | 0.70 | 51.34 | 0.70 |
| + Elastic Attention (FA-SSA) | **46.15** | 0.64 | 46.54 | 0.65 | 28.19 | 0.72 | **67.52** | 0.71 | 48.07 | 0.65 | **62.95** | 0.78 | 51.51 | 0.69 |
| + Elastic Attention (FA-XA) | 44.01 | 0.75 | **49.99** | 0.76 | 28.30 | 0.83 | 66.23 | 0.80 | **50.95** | 0.77 | 60.57 | 0.86 | **51.66** | 0.80 |
| Llama-3.1-8B-Instruct backbone model | | | | | | | | | | | | | | |
| Llama-3.1-8B-Instruct | 48.75 | - | 51.85 | - | 30.26 | - | 68.16 | - | 56.00 | - | 55.81 | - | 53.28 | - |
| + InfLLM-V2 | 43.77 | - | 46.30 | - | 30.08 | - | 67.32 | - | 42.03 | - | **64.30** | - | 50.73 | - |
| + DuoAttention | 48.65 | 0.70 | 45.21 | 0.70 | 29.93 | 0.70 | 67.35 | 0.70 | **54.56** | 0.70 | 56.47 | 0.70 | 51.82 | 0.70 |
| + PruLong | 47.69 | 0.70 | 43.05 | 0.70 | 30.02 | 0.70 | 67.86 | 0.70 | 53.56 | 0.70 | 61.07 | 0.70 | 52.11 | 0.70 |
| + Elastic Attention (FA-SSA) | **49.92** | 0.64 | 48.92 | 0.63 | 30.14 | 0.73 | 67.99 | 0.74 | 54.00 | 0.64 | 60.70 | 0.79 | **53.35** | 0.69 |
| + Elastic Attention (FA-XA) | 49.40 | 0.71 | **52.94** | 0.69 | **30.30** | 0.80 | **68.55** | 0.79 | 49.66 | 0.72 | 56.49 | 0.87 | 52.71 | 0.77 |

et al., 2025), which decouple the sparsity–performance trade-offs across tasks and mitigate optimization conflicts.

### 3.3. Efficient Deployment

Since each layer adopts a hybrid-heads computation, where different types of heads cannot be efficiently parallelized at runtime, we implement a fused attention kernel that jointly computes retrieval heads and sparse heads "simultaneously". In this work, we focus on single-GPU deployment, without introducing inter-device communication overhead, e.g., head-wise parallelism. In practice, attention kernels are launched over a three-dimensional grid spanning the sequence dimension, attention heads, and batch. When the input sequence length is sufficiently large, parallelism along the sequence dimension dominates the overall execution. Since the number of concurrently active blocks is bounded by the available streaming multiprocessors, the GPU effectively schedules and executes almost all sequence blocks of one head before progressing to the next head. We report the speedup over a Torch-based sequential, layer-wise hybrid attention implementation in Figure 4, which indicates that our fused kernel yields prefill-time acceleration during inference. We show more details in Appendix D.

## 4. Experiment

### 4.1. Setting

**Training and Data**   We select Qwen3-(4B/8B) (Yang et al., 2025) and Llama-3.1-8B-Instruct (Grattafiori et al.,

2024) as the backbone LLMs, and build training dataset by combining five sources: ChatQA2-Long-SFT-data (Xu et al., 2024), MuSiQue (Trivedi et al., 2022), CoLT-132K (Li et al., 2025), GovReport (Huang et al., 2021), and XSum (Narayan et al., 2018), covering both sparsity-sensitive tasks (Single-Doc QA and Multihop QA) and sparsity-robust tasks (code completion, summarization, and in-context learning). The resulting dataset spans sequence lengths ranging from 8K to 64K tokens, comprising approximately 0.74B tokens in total. For sparsity-robust and sparsity-sensitive task categories, we empirically set the $t = 1.0$ and $t = 0.7$, respectively. We conduct training with $8 \times$A800 GPUs, with each run completing within 12 hours. We provide more training details in Appendix C and list hyperparameters in Table 8.

**Evaluation**   We compare our method against representative training-based sparsity approaches, including DuoAttention (Xiao et al., 2025), PruLong (Bhaskar et al., 2025), and InfLLM-V2 (Zhao et al., 2025). For the attention computation mode of sparse heads, we consider Streaming Sparse Attention (SSA) (Xiao et al., 2024b) and XAttention (XA) (Xu et al., 2025). For XA, we set the threshold $\tau = 0.9$, while all other hyperparameters are kept at their default values. We denote different head computation configurations using the "{Retrieval Head mode}–{Sparse Head mode}" notation, e.g., FA-SSA means retrieval heads use FA mode and sparse heads use SSA mode. Due to space constraints, experiments on MoBA (Lu et al., 2025a) and NSA (Yuan et al., 2025), along with detailed baseline configurations, are shown in Appendix F. All the evaluations are conducted

*Table 2.* Model performance on RULER (Hsieh et al., 2024) and LongBench-v2 (Bai et al., 2025). We report the average Perf. and $\Omega_{\mathrm{MSR}}$.

| Models | RULER | | | | | | | | LongBench-v2 | | | |
|---|---|---|---|---|---|---|---|---|---|---|---|---|
| | 8K | 16K | 32K | 64K | 128K | 256K | Perf. | $\Omega_{\mathrm{MSR}}$ | Easy | Hard | Perf. | $\Omega_{\mathrm{MSR}}$ |
| **Qwen3-4B backbone model** | | | | | | | | | | | | |
| Qwen3-4B | 87.49 | 86.82 | 60.05 | 70.98 | 53.19 | 43.27 | 66.00 | - | 32.67 | 22.18 | 25.96 | - |
| + InfLLM-V2 | 78.59 | 74.40 | 43.39 | 44.94 | 28.57 | 27.01 | 51.02 | - | 28.00 | 24.06 | 25.48 | - |
| + DuoAttention | 76.82 | 75.17 | 50.53 | 61.42 | 45.86 | 45.46 | 58.30 | 0.70 | 28.67 | 24.44 | 25.96 | 0.70 |
| + PruLong | 77.03 | 73.99 | 51.32 | 60.98 | 42.70 | **48.76** | 58.38 | 0.70 | 27.60 | 22.33 | 24.35 | 0.70 |
| + Elastic Attention (FA-SSA) | 83.35 | 79.24 | 50.79 | 67.03 | 47.83 | 47.32 | 61.81 | 0.66 | **34.00** | 24.44 | 27.88 | 0.70 |
| + Elastic Attention (FA-XA) | **86.56** | **85.38** | 56.88 | 69.42 | 49.48 | 43.47 | **63.27** | 0.67 | 32.00 | **25.94** | **28.12** | 0.72 |
| **Qwen3-8B backbone model** | | | | | | | | | | | | |
| Qwen3-8B | 89.69 | 85.62 | 63.23 | 82.39 | 65.84 | 66.71 | 75.74 | - | 39.33 | 27.82 | 31.97 | - |
| + InfLLM-V2 | 77.01 | 68.53 | 25.93 | 53.47 | 34.96 | 32.95 | 52.58 | - | 29.32 | 28.67 | 29.09 | - |
| + DuoAttention | 81.03 | 78.85 | 56.61 | 70.10 | 54.68 | 56.28 | 65.94 | 0.70 | 40.67 | 25.56 | 31.01 | 0.70 |
| + PruLong | **87.20** | 82.69 | 61.64 | 73.05 | 59.74 | 58.82 | 69.90 | 0.70 | 38.67 | 27.82 | 31.73 | 0.70 |
| + Elastic Attention (FA-SSA) | 86.62 | 82.81 | 64.55 | 77.41 | 61.17 | 61.75 | 71.74 | 0.65 | 37.33 | **31.20** | **33.41** | 0.66 |
| + Elastic Attention (FA-XA) | 85.07 | **85.12** | 65.08 | 82.34 | 64.57 | 63.41 | **73.87** | 0.76 | 30.68 | 30.77 | 30.74 | 0.78 |
| **Llama-3.1-8B-Instruct backbone model** | | | | | | | | | | | | |
| Llama-3.1-8B-Instruct | 92.88 | 92.83 | 89.46 | 70.79 | 80.12 | 72.34 | 83.47 | - | 32.00 | 33.08 | 32.69 | - |
| + InfLLM-V2 | 89.30 | 80.93 | 60.98 | 35.90 | 32.29 | 47.27 | 59.10 | - | 29.32 | 28.67 | 29.09 | - |
| + DuoAttention | 87.20 | 77.85 | 66.99 | 40.13 | 57.45 | 42.92 | 62.92 | 0.70 | 28.67 | 27.44 | 27.88 | 0.70 |
| + PruLong | 83.54 | 69.35 | 56.83 | 30.88 | 33.74 | 21.64 | 48.82 | 0.70 | 28.00 | 28.92 | 28.61 | 0.70 |
| + Elastic Attention (FA-SSA) | 89.93 | 83.42 | 80.20 | 56.30 | 68.16 | 56.47 | 72.85 | 0.65 | 28.00 | 29.70 | 29.09 | 0.68 |
| + Elastic Attention (FA-XA) | **92.82** | **92.00** | **87.80** | 68.23 | 78.87 | 68.51 | **81.82** | 0.72 | 30.67 | 30.83 | 30.77 | 0.75 |

with `LOOM-Eval` evaluation framework (Tang et al., 2025).

### 4.2. Evaluation Results

**Real-world Long-context Tasks** We first evaluate our method on LongBench-E (Bai et al., 2024), a real-world long-context benchmark comprising 14 tasks across 6 categories with varying context lengths. We compare Elastic Attention with other baselines in Table 1. Within each comparison group, all methods share the same backbone model and are trained with the same data. We can observe that our method consistently achieves the best average performance, overcoming or approaching the backbone (fully FA) while enabling more efficient inference. Across different tasks, Elastic Attention can dynamically allocate different sparsity levels, e.g., achieving an average $\Omega_{\mathrm{MSR}}$ of around 0.85 on code tasks and 0.68 on QA. Notably, we find that our method underperforms some baselines on sparsity-robust tasks (e.g., Code and Summ). This stems from our higher sparsity and InfLLM-V2's usage of FA in specific cases[6].

**Length Extrapolation Capability Testing** We evaluate our method on RULER (Hsieh et al., 2024), aiming to examine model length extrapolation performance across different context length regimes. Notably, the maximum training context length is 64K tokens, and we evaluate models on lengths

---

[6]InfLLM-V2 does not have a well-defined $\Omega_{\mathrm{MSR}}$, as it employs FA when the context length is below 8K tokens and switches to a hybrid attention computation mode for longer contexts.

*Table 3.* Performance comparison across different benchmarks. The best results in each column are highlighted in **bold**. The values in parentheses indicate the performance gap relative to the Qwen3-4B baseline.

| Model | AIME24 | GSM8K | Math | LongHealth | AVG |
|---|---|---|---|---|---|
| Qwen3-4B | 6.70 | 43.10 | 55.80 | 63.92 | 42.38 |
| DuoAttention | 6.70 | **45.80** | 52.40 | 49.98 | 38.72 (-3.66) |
| Elastic Attention (FA-SSA) | **10.00** | **45.80** | **57.10** | 59.42 | 43.08 (+0.70) |
| Elastic Attention (FA-XA) | 7.10 | 45.60 | 56.30 | **64.40** | 43.35 (+0.97) |

ranging from 8K to 256K tokens. As shown in Table 2 (left group), our approach achieves the best performance across all evaluated context lengths while maintaining favorable sparsity levels, with $\Omega_{\mathrm{MSR}}$ values consistently in the range of $0.65 \sim 0.7$. Besides, we observe that for 8B-scale models at context lengths exceeding 64K tokens, the FA–XA configuration consistently yields the best performance. This is because FA–XA attains a lower $\Omega_{\mathrm{ESR}}$ than the other configurations, allowing it to retain and exploit more relevant information. This advantage becomes particularly pronounced at extremely long contexts, e.g., at 256K tokens, FA–XA still maintains strong performance.

**Long-form Reasoning Task** We further evaluate our models on LongBench-V2 (Bai et al., 2025), a widely recognized long-context reasoning benchmark with context lengths ranging from 8K to 2M words. As shown in Table 2 (right group), our approach consistently delivers strong results across both Easy and Hard settings among all compared

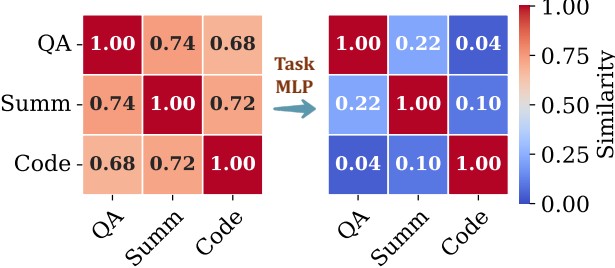

*Figure 5.* Visualization of task representation similarity. (Left) before Task MLP, the pooled hidden states exhibit high pairwise cosine similarity across different tasks; (Right) after passing through the Task MLP, the inter-task similarity significantly decreases.

*Table 4.* Comparison among different MLP hidden dimensions.

| Hidden Size | LongBench | RULER | LongBench-V2 | Avg. |
|---|---|---|---|---|
| $2 \times d'$ | 48.07 | 60.67 | 26.92 | 45.22 |
| $4 \times d'$ (default) | 48.08 | 61.81 | 27.88 | 45.92 |
| $6 \times d'$ | 48.07 | 62.08 | 26.20 | 45.45 |
| $8 \times d'$ | 48.47 | 65.27 | 25.48 | 46.40 |

methods, as well as achieves the best average performance (either FA-SSA or FA-XA setting). Interestingly, we observe that the FA–XA configuration does not yield strong performance on Qwen3-8B, which may be attributed to model-specific characteristics of this model and could benefit from more fine-grained hyperparameter tuning.

**Real-world Generation and Domain-specific Understanding** To test our approach in practical settings, we evaluate it on complex mathematical reasoning (AIME24 (MAA, 2024), GSM8K (Cobbe et al., 2021), and Math (Hendrycks et al., 2021)) and domain-specific long-context understanding (LongHealth (Adams et al., 2024)). As Table 3 shows, our method preserves and occasionally improves the performance of the Qwen3-4B model. The FA-SSA configuration achieves the highest scores across all mathematical reasoning benchmarks, raising the AIME24 score from 6.70 to 10.00. The FA-XA variant excels in domain-specific tasks, reaching 64.40 on LongHealth and obtaining the highest overall average of 43.35 (+0.97 over the baseline). DuoAttention, however, shows an average performance drop of 3.66, largely due to a severe decline on the LongHealth benchmark.

## 5. Ablation Study

In this section, we analyze the Elastic Attention preliminary on the Qwen3-4B and Llama3.1-8B-Instruct models. We investigate the design of its core component, i.e., the Attention Router module (§ 5.1); study the impact of different target sparsity $t$ (§ 5.2); analyze our method's trade-off between performance and inference efficiency (§ 5.3); and show the scalability of Elastic Attention (§ 5.4).

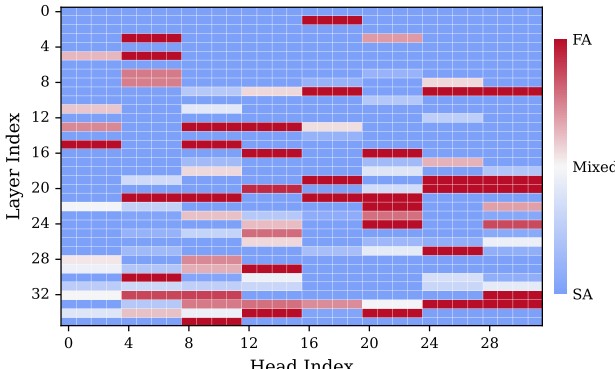

*Figure 6.* Overview of routing activation frequency of each head in Qwen3-4B. Red indicates heads that are consistently routed to FA (i.e., retrieval heads) across all 6 tasks in LongBench-E, while blue denotes heads that are consistently routed to SA.

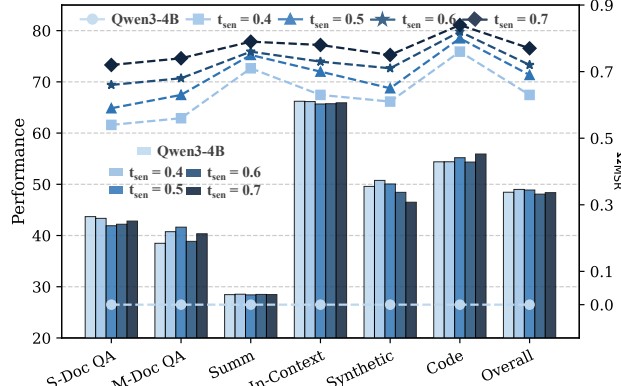

*Figure 7.* Comparison of performance and test-time $\Omega_{\mathrm{MSR}}$ among different training sparsity target $t$ settings. The bar chart denotes the performance and the line chart denotes $\Omega_{\mathrm{MSR}}$ in each task.

### 5.1. Design of Attention Router

**Effect of Task MLP** We analyze the role of the Task MLP within the Attention Router. Specifically, we take the Task MLP from the first layer of the model and calculate the pairwise cosine similarity of task-specific hidden states ($x_K$) **before and after** they are processed by the Task MLP. Notably, a lower cosine similarity indicates greater separation between task representations, reflecting improved task discrimination. As shown in Figure 5, we compare representations across three tasks and observe that, after passing through the Task MLP, inter-task similarity is significantly reduced. This result indicates that the Task MLP effectively disentangles task-specific features and produces more discriminative representations for the subsequent routing decisions performed by the Router MLP. We provide more details and results of other models in Appendix G.1.

**Impact of MLP Intermediate Hidden Size** We study the impact of the intermediate hidden size in the Task- and the Router-MLP layers within the Attention Router. Specifi-

cally, we experiment with the intermediate dimension from $\times 2 \sim \times 8$, where $\times 4$ corresponds to the default setting used in our main experiments. As shown in Table 4, all settings achieve very similar average performance across 3 benchmarks. Although the $\times 8$ setting attains the highest overall score, we adopt the $\times 4$ setting in our main experiments, as it provides a more favorable trade-off between performance gains and additional parameter overhead.

**Attention Mode Routing Analysis** The attention computation modes assigned by the Attention Router exhibit strong and consistent patterns at the head level. We observe that, in LLMs, certain heads primarily leverage FA computation mode, while others are frequently assigned with SA. As shown in Figure 6, we analyze the distribution of attention computation modes for each head across the 6 LongBench-E sub-tasks. We find that a subset of heads, which are predominantly located in the middle to higher layers, are consistently activated in FA mode. This observation is consistent with prior findings in Wu et al. (2024), indicating that these heads function as retrieval heads. Partial heads that are highlighted in lighter colors exhibit switching behavior across tasks, and the remaining heads are consistently routed to SA. More results are shown in Appendix G.2.

## 5.2. Impact of Target Sparsity $t$ Allocation

We study the impact of target sparsity $t$ (Eq. 8) to model performance. Specifically, we fix the target sparsity of sparsity-robust tasks ($t_{rob}$) to 1, while progressively decreasing target sparsity for sparsity-sensitive tasks ($t_{sen}$) from 0.7 to 0.4. As shown in Figure 7, as $t_{sen}$ decreases, the resulting ($\Omega_{MSR}$) allocated by the model exhibits slightly greater task-level differentiation across different tasks. Yet, we can find that $\Omega_{MSR}$ does not strictly match the target $t$. This is because we adopt *task-dependent and non-tight constraints*, which do not force the model to exactly satisfy the prescribed sparsity. We provide full training curves with explanations in Appendix G.4. Besides, when $t_{sen}$ is set too low (e.g., 0.4), i.e., allocating a higher proportion of FA computation, the overall performance can even surpass that of the backbone. Yet, from an inference-efficiency perspective, we adopt $t_{sen} = 0.7$ and $t_{rob} = 1.0$ in our main experiments, which achieves a favorable balance between strong overall performance and high inference efficiency.

## 5.3. Overall Performance and Inference Efficiency

We compare different methods on RULER across multiple context-length regimes in terms of both task performance and inference efficiency. Implementation details are provided in Appendix C.2. As shown in Figure 8, our method consistently achieves the best performance among all compared approaches. In addition, its inference speedup increases with longer context lengths, as the model dynami-

*Table 5.* Results of implementing retrieval heads with XA.

| Method | LongBench | RULER | LongBench-V2 | Avg. |
|---|---|---|---|---|
| Qwen3-4B | 48.45 | 66.00 | 25.96 | 46.80 |
| + XA-SSA | 48.14 | 63.70 | 25.96 | 45.93 |
| Qwen3-8B | 52.16 | 75.74 | 31.97 | 53.26 |
| + XA-SSA | 49.25 | 66.31 | 32.69 | 49.42 |
| Llama-3.1-8B | 53.28 | 83.47 | 32.69 | 56.48 |
| + XA-SSA | 50.71 | 70.27 | 29.09 | 50.02 |

cally allocates higher $\Omega_{MSR}$ for longer inputs. In contrast, several baselines are constrained by architectural designs. For example, NSA and InfLLM-V2 impose strict divisibility requirements on the number of KV attention heads (e.g., multiples of 16), which are not well aligned with Llama-3.1-8B. Moreover, MoBA and InfLLM-V2 must reserve part of the computation budget to pre-compute sequence-level features, leading to GPU out-of-memory failures at 256K context length. Finally, we analyze $\Omega_{ESR}$, computed based on the actual number of attended tokens. We observe that, as sequence length increases, training-based hybrid models maintain an approximately constant $\Omega_{ESR}$, whereas our method achieves a consistently lower $\Omega_{ESR}$ and slightly outperforms comparable baselines (e.g., PruLong), indicating superior scalability in effective sparsification. We show more details and comparison results in Appendix G.3.

*Table 6.* Performance comparison of different tuning methods on LongBench and RULER. The best results are **bolded**, and the second-best are underlined.

| Model | LongBench | RULER |
|---|---|---|
| Qwen3-4B | 48.45 | **66.00** |
| Elastic Attention (FA-SSA) | 48.08 | 61.81 |
| Elastic Attention (FA-SSA) LoRA | 49.05 | 61.62 |
| Elastic Attention (FA-SSA) Full | **50.40** | 61.97 |

## 5.4. Scalability of Elastic Attention

We assess the scalability of Elastic Attention by implementing retrieval heads with XA, yielding the XA–SSA setting where the entire model operates under a full SA regime. As shown in Table 5, the XA–SSA setting can still preserve strong performance. For instance, for Qwen3-4B, the average performance gap across three long-context benchmarks even remains within 1 point. Although 8B-scale models exhibit performance degradation, this trade-off is accompanied by a substantial improvement in inference speed, owing to the fully sparse attention structure of the model.

To test elastic sparsity beyond the standard frozen-backbone paradigm, we explore the impact of continued pretraining. Although training from scratch contradicts our mo-

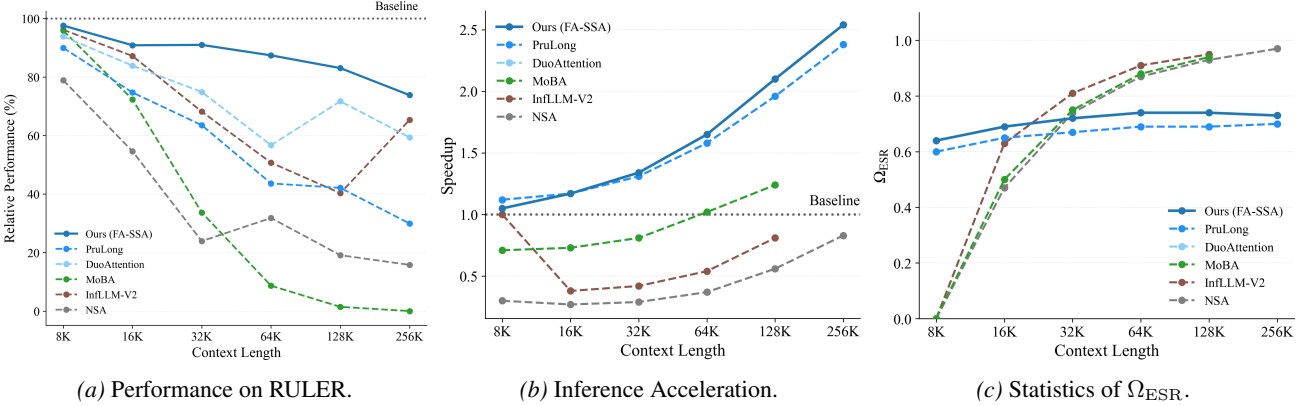

*(a)* Performance on RULER.    *(b)* Inference Acceleration.    *(c)* Statistics of $\Omega_{\text{ESR}}$.

*Figure 8.* Comparison of performance and inference speedup on the RULER benchmark across different methods. We adopt Llama-3.1-8B-Instruct as the backbone model and compare with training-based methods (FA–SSA), as well as other cutting-edge sparse attention methods. We report $\Omega_{\text{ESR}}$, as it provides a fair comparison of the effective proportion of attended tokens across different approaches.

tivation since pre-trained models already contain functional retrieval heads, we relax the frozen backbone constraint using LoRA (Hu et al., 2022) and full fine-tuning. Updating more parameters improves model performance on Long-Bench(as shown in Table 6). Meanwhile, synthetic long-context retrieval performance matches the frozen-backbone results reported earlier. This indicates that Elastic Attention integrates smoothly with continued pretraining, improving general task performance without sacrificing existing retrieval skills.

## 6. Conclusion

We proposed *Elastic Attention*, a test-time adaptive sparse attention that dynamically adjusts model sparsity based on the input. We show that effective sparsity allocation can be achieved by distinguishing between sparsity-robust and sparsity-sensitive task regimes. Thus, Elastic Attention works by introducing a lightweight Attention Router that performs head-wise routing between FA and SA modes based on the input task regimes. Notably, Elastic Attention brings negligible overhead, eliminating the modification of pretrained backbones. Experiments across 3 widely accepted long-context benchmarks on different cutting-edge LLMs demonstrate the superiority of our methods.

## Impact Statement

In this paper, we introduce **Elastic Attention**, a test-time adaptive mechanism that dynamically allocates sparsity for hybrid attention models based on the input. We summarize the broader impact of our method from two perspectives.

(1) Our approach is particularly well-suited for small- to medium-scale LLMs that can be deployed on a single GPU or a limited number of devices. In contrast, extremely large-scale LLMs are often deployed with

head-level parallelism in practical inference systems, which may conflict with layer-wise hybrid attention designs such as ours (Zhang et al., 2025a). For end users with long input sequences, such as on-device agent models with extended reasoning trajectories (Luo et al., 2025) or document understanding (Chia et al., 2025) workloads, Elastic Attention substantially reduces computational overhead and inference latency, while automatically balancing model quality and efficiency through input-dependent sparsity allocation.

(2) More broadly, we hope this work motivates the exploration of next-generation model architectures that go beyond sparse attention and elevate dynamic, input-dependent computation to a first-class design principle, such as Titans (Behrouz et al., 2024). As task complexity and contextual demands vary widely, future models should increasingly adapt their computational behavior to individual inputs, rather than adhering to a rigid, fully static execution (inference) paradigm.

## Acknowledge

We want to thank all the anonymous reviewers for their valuable comments. This work was supported by the National Science Foundation of China (NSFC No. 62576232) and sponsored by CCF-Baidu Open Fund.

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

# A. Related Work

## A.1. Sparse Attention Mechanisms

To mitigate the quadratic complexity of standard attention, existing research has broadly evolved along two trajectories: inference-time heuristics and training-aware sparsification. Inference-time heuristics typically employ static patterns, such as fixed sliding windows or strides (Xiao et al., 2024b; He et al., 2025; Beltagy et al., 2020), to restrict the receptive field. To capture more dynamic dependencies, content-aware approaches have been proposed. Token eviction policies discard uninformative tokens based on accumulated importance scores (Zhang et al., 2023; Li et al., 2024; Liu et al., 2024), while kernel-based estimators identify salient blocks to bypass redundant computations (Jiang et al., 2024). Complementarily, prefill optimizers leverage importance-driven selection to accelerate long-context processing (Lai et al., 2025; Xu et al., 2025; Zhang et al., 2025b; Peng et al., 2025b; Ji et al., 2025). Despite their effectiveness, these heuristic methods often hinge on sensitive hyperparameters, limiting their robustness across varying tasks.

In contrast, internalize sparsity within the optimization objective to align training with sparse inference. A primary direction involves learnable selection. For instance, SeerAttention (Gao et al., 2024), NSA (Yuan et al., 2025) and MoBA (Lu et al., 2025a) employ learnable gates and hierarchical constraints to approximate ground-truth attention patterns. To bridge the gap between dense pre-training and sparse adaptation, InfLLM-v2 (Zhao et al., 2025) introduces a dense-sparse switchable mechanism via parameter-free pooling, while DSA (DeepSeek-AI, 2025) utilizes a lightning indexer with a two-stage training strategy to efficiently filter top-$k$ key-value pairs. However, most of these methods focus on fine-grained block-level selection within a fixed attention mechanism, rather than dynamically adapting the attention mode itself based on input complexity.

## A.2. Hybrid Efficient Architectures

To balance efficiency with performance, hybrid architectures strategically integrate Full Attention (FA) with linear-complexity operators. The dominant paradigm, inter-layer hybridization, interleaves linear layers (e.g., SSMs or RNNs) with standard attention layers to recover associative recall capabilities (Ku et al., 2025; Dao & Gu, 2024). Notable large-scale implementations, such as Jamba (Lieber et al., 2024), utilize fixed block-wise ratios, while variants optimize memory via shared global blocks (Glorioso et al., 2024) or sliding windows (Ren et al., 2024).

More recently, intra-layer hybridization has emerged to refine granularity. For example, PruLong (Bhaskar et al., 2025) and DuoAttention (Xiao et al., 2024a) combine FA and Streaming Sparse Attention (SSA) within individual layers, assigning different heads to different modes. LongCat (Zhang et al., 2025a) proposes the LoZA mechanism, constructing a static ZigZag topology by replacing low-sensitivity Multi-head Latent Attention (MLA) modules with linear-complexity SSA. Nevertheless, a critical limitation of these approaches is their reliance on static topologies or pre-defined ratios determined prior to inference. Such rigid designs lack the flexibility to distinguish between sparsity-robust and sparsity-sensitive tasks dynamically, often leading to suboptimal resource allocation for diverse inputs.

# B. Retrieval Scoring and Sparsification Setup

In this section, we provide a detailed description of the retrieval head identification and the progressive sparsification strategy mentioned in Section 2.2.

## B.1. Retrieval Score Calculation

Following the methodology proposed by Retrieval Head (Wu et al., 2024), we identify and rank attention heads based on their ability to retrieve specific information from long contexts. We employ the Needle-in-a-Haystack probing method using the Llama-3.1-8B-Instruct (Grattafiori et al., 2024) model, where specific key information (the "needle") is inserted into a long context (the "haystack").

For a given attention head $h$ in layer $\ell$, denoted as $H_{\ell,h}$, we calculate its Retrieval Score ($S_{\ell,h}$) by measuring the attention mass allocated to the needle tokens. Formally, let $s$ be the input sequence length, $\mathcal{O}^{(\ell,h)} \in \mathbb{R}^{s \times s}$ be the attention matrix, and $\mathcal{I}_{needle}$ be the set of indices corresponding to the needle tokens. The score is computed as:

$$S_{\ell,h} = \frac{1}{|\mathcal{D}|} \sum_{x \in \mathcal{D}} \sum_{j \in \mathcal{I}_{needle}} \mathcal{O}^{(\ell,h)}_{T,j} \tag{9}$$

*Table 7.* Performance retention rates across various model sparsity ratios. The values represent the percentage of performance relative to the Full Attention baseline (Sparsity 0.0), where 100.00 indicates parity. Rows denote the model sparsity ratio, and columns denote the evaluation tasks.

| Sparsity | Single-Doc QA | Multi-Hop QA | Summarization | Few-Shot | Synthetic | Code |
|---|---|---|---|---|---|---|
| Full ($\Omega_{\text{MSR}} = 0.0$) | 100.00 | 100.00 | 100.00 | 100.00 | 100.00 | 100.00 |
| $\Omega_{\text{MSR}} = 0.1$ | 85.37 | 96.23 | 99.38 | 96.54 | 92.27 | 99.07 |
| $\Omega_{\text{MSR}} = 0.2$ | 71.32 | 73.84 | 98.00 | 93.83 | 81.26 | 98.49 |
| $\Omega_{\text{MSR}} = 0.3$ | 61.94 | 69.07 | 96.19 | 91.83 | 59.28 | 98.10 |
| $\Omega_{\text{MSR}} = 0.4$ | 59.36 | 66.48 | 94.75 | 87.01 | 47.61 | 95.58 |
| $\Omega_{\text{MSR}} = 0.5$ | 57.95 | 64.33 | 93.02 | 87.33 | 45.31 | 99.54 |
| $\Omega_{\text{MSR}} = 0.6$ | 56.95 | 59.66 | 94.68 | 87.05 | 44.75 | 98.66 |
| $\Omega_{\text{MSR}} = 0.7$ | 56.08 | 61.91 | 92.52 | 88.51 | 41.25 | 97.73 |
| $\Omega_{\text{MSR}} = 0.8$ | 56.04 | 59.55 | 93.73 | 89.07 | 47.98 | 99.86 |
| $\Omega_{\text{MSR}} = 0.9$ | 56.02 | 63.40 | 94.00 | 90.19 | 48.37 | 98.48 |
| $\Omega_{\text{MSR}} = 1.0$ | 56.46 | 61.45 | 93.42 | 90.19 | 47.98 | 99.46 |

where $\mathcal{D}$ represents the validation dataset, and $\mathcal{O}_{T,j}^{(\ell,h)}$ denotes the attention weight from the last token (query position $T$) to the needle position $j$. A higher $S_{\ell,h}$ indicates that the head frequently and strongly activates on relevant information in long contexts.

### B.2. Progressive Sparsification Strategy

Based on the computed scores $S_{\ell,h}$, we rank all $L \cdot H$ attention heads across the model in descending order. As defined in the main text, the **Model Sparsity Ratio** ($\Omega_{\text{MSR}}$) represents the proportion of heads converted to local attention.

To simulate the varying levels of sparsity reported in our experiments (e.g., $\Omega_{\text{MSR}} = 20\%$), we employ a thresholding mechanism:

1. We determine the number of heads to preserve as Full Attention (FA) via $k = \lfloor (1 - \Omega_{\text{MSR}}) \cdot (L \cdot H) \rfloor$.

2. The top-$k$ heads with the highest retrieval scores are retained as **Retrieval Heads** (keeping FA) to ensure global information integration.

3. The remaining heads (the bottom ranked ones) are replaced with **Streaming Sparse Attention (SSA)** heads.

## C. Implementation Details

In this section, we provide a comprehensive overview of the training configurations, baseline implementation details, and system-level optimizations tailored for efficient long-context processing.

### C.1. Training Configuration and Hyperparameters

**Model Architecture.** We validate the scalability of our approach across diverse model sizes, ranging from lightweight architectures like Qwen3-4B to widely-adopted mid-class models such as Qwen3-8B (Yang et al., 2025) and Meta-Llama-3.1-Instruct (Grattafiori et al., 2024). To preserve the pre-trained model's general capabilities, we adhere to a parameter-efficient fine-tuning protocol: the pre-trained backbone is frozen, and only the *Attention Router* parameters are optimized. Regarding task representation, we employ a boundary-pooling strategy that exclusively aggregates the first 100 and last 100 tokens of the sequence. These segments are selected as they typically encapsulate critical system instructions and user queries essential for accurate task identification.

**Optimization Setup.** All models are trained with a sequence length of $L = 65,536$ tokens using 'bfloat16' precision to accommodate long-context dependencies. The training utilizes the AdamW optimizer (Loshchilov & Hutter, 2019) ($\beta_1 = 0.9, \beta_2 = 0.95$) on a distributed cluster employing Fully Sharded Data Parallel (FSDP) with a Hybrid Sharding strategy. We adopt a decoupled learning rate strategy to balance router convergence and sparsity regularization:

*Table 8.* Hyperparameters: General configuration (Left) and Baseline-specific settings (Right).

*(a)* General Config

| Hyperparameter | Value |
|---|---|
| *Model & Training* | |
| Base Model | Qwen, Llama |
| Sequence length | 65536 |
| Precision | bfloat16 |
| Global Batch Size | 48 |
| Training Steps | 300 |
| Mask / Reg. LR | $5e^{-4}$ / $1e^{-3}$ |
| Warmup Ratio | 0.2 |
| AdamW Momentum ($\beta_1, \beta_2$) | (0.9, 0.95) |
| Weight Decay | 0.1 |
| Learning Rate Schedule | Cosine |
| *Sparsity Config* | |
| Sink / Local Size | 128 / 2048 |
| Block / Chunk Size | 64 / 16384 |
| Stride / Threshold | 16 / 0.9 |
| Selection Mode | Inverse |

*(b)* Baseline-Specifics

| Param | MoBA | NSA | InfLLMv2 |
|---|---|---|---|
| *Structure & Kernel* | | | |
| Block Size | 1024 | 64 | 64 |
| Top-$k$ | 8 | 128 | 64 |
| Window Size | - | 512 | 2048 |
| Kernel Size | - | 32 | 32 |
| Kernel Stride | - | 16 | 16 |
| *Learnable Params* | | | |
| Q/K/V Proj | ✓ | ✓ | ✓ |
| Gate | - | ✓ | - |
| Compress K/V | - | ✓ | - |
| *Extra Config* | | | |
| Compress Type | Pooling | Linear | Pooling |
| Use NoPE | - | - | False |
| Dense Len | - | - | 8192 |

- **Router Parameters:** A learning rate of $5 \times 10^{-4}$ is applied to the attention router to facilitate rapid adaptation to retrieval patterns.

- **Regularization Coefficients:** A higher learning rate of $1 \times 10^{-3}$ is assigned to the sparsity regularization terms. **Specifically, the dual regularization coefficients $\lambda_1$ and $\lambda_2$ are randomly initialized and optimized alongside the router parameters.**

We utilize a cosine decay learning rate schedule following a linear warmup phase spanning the first 20% of total training steps.

### C.2. Baseline Implementation Details

To rigorously evaluate the efficacy of our approach, we benchmark against a comprehensive suite of state-of-the-art sparse attention mechanisms. These are categorized into training-free methods and training-based adaptation methods.

**Training-Free Methods.** We employ XAttention [7] (Xu et al., 2025) as the primary training-free baseline. This category relies on heuristic-based sparsity without parameter updates.

**Training-Based Methods.** For methods requiring training adaptation, including InfLLM_v2 [8] (Zhao et al., 2025), MoBA [9] (Lu et al., 2025a), NSA [10] (Yuan et al., 2025), PruLong [11] (Bhaskar et al., 2025), and DuoAttention [12] (Xiao et al., 2025), we implement a unified fine-tuning protocol to ensure strict fairness. Unlike our method, which freezes the backbone entirely, most competing methods require updating projection layers. For InfLLM_v2, MoBA, and NSA, we restrict the trainable scope to the query-key-value projection weights ($W_{qkv}$) and their respective method-specific parameters, freezing the remaining backbone. All baselines are trained under the same environment and dataset, and the hyperparameters are strictly adhered to their original setups, such as a block size of 1024 for MoBA versus 64 for NSA, and the specific dense context length (8K) for InfLLM_v2. Detailed hyperparameter comparisons are provided in Table 8 (Right).

---

[7] https://github.com/mit-han-lab/x-attention
[8] https://modelscope.cn/models/OpenBMB/MiniCPM4-8B/file/view/master/modeling_minicpm.py
[9] https://github.com/MoonshotAI/MoBA
[10] https://github.com/XunhaoLai/native-sparse-attention-triton
[11] https://github.com/princeton-pli/PruLong
[12] https://github.com/mit-han-lab/duo-attention

**Algorithm 1** Comparison of Serial Dispatch (Baseline) vs. Parallel BSA (Ours)

| (a) PyTorch Baseline | (b) Ours (Parallel via BSA) |
|---|---|
| **Input:** $\mathbf{Q}, \mathbf{K}, \mathbf{V}$, Router $\mathcal{R}$ | **Input:** $\mathbf{Q}, \mathbf{K}, \mathbf{V}$, Router $\mathcal{R}$ |
| 1: $\mathbf{r} \leftarrow \mathcal{R}(\mathbf{x})$ | 1: $\mathbf{r} \leftarrow \mathcal{R}(\mathbf{x})$ |
| | 2: $\mathbf{m} \leftarrow \text{Map}(\mathbf{r})$ |

(a) PyTorch Baseline — line 2:

**Step 1: Serial Split**
$\mathcal{I}_{\text{full}} \leftarrow \{h \mid \mathbf{r}_h = 0\}$
$\mathcal{I}_{\text{sp}} \leftarrow \{h \mid \mathbf{r}_h = 1\}$
$\mathbf{Q}_{\text{full}} \leftarrow \mathbf{Q}[:, \mathcal{I}_{\text{full}}]$

**Step 2: Separate Comp.**
$\mathbf{O}_{\text{full}} \leftarrow \text{FlashAttn}(\ldots)$
$\mathbf{O}_{\text{sp}} \leftarrow \text{SlidingWin}(\ldots)$

**Step 3: Merge Results**
$\mathbf{O}[:, \mathcal{I}_{\text{full}}] \leftarrow \mathbf{O}_{\text{full}}$
$\mathbf{O}[:, \mathcal{I}_{\text{sp}}] \leftarrow \mathbf{O}_{\text{sp}}$

(b) Ours (Parallel via BSA) — line 3:

**Step 1: Unified Execution**
$\mathbf{O} \leftarrow \text{BSA\_Krn}(\mathbf{Q}, \mathbf{K}, \mathbf{V}, \mathbf{m})$

*Inside Kernel:*
**par_for** $h$ **do**
  **if** $\mathbf{m}[h] == \text{SP}$ **then**
    $\mathbf{O}[h] \leftarrow \text{Sparse}(\ldots);$
  **else**
    $\mathbf{O}[h] \leftarrow \text{Full}(\ldots);$
  **end if**
**end par_for**

## C.3. Sparsity and Kernel Configuration

To achieve efficient streaming inference, we employ Block-Sparse-Attention (Guo et al., 2024). This configuration governs the granularity and retention policy of the attention mechanism:

- **Block Size:** Set to 64, defining the minimum unit of sparsity.

- **Chunk Size:** Set to 16,384, enabling the processing of ultra-long sequences.

- **Sink Token Strategy:** We enforce a "sink token" size of 128 to preserve the attention sink phenomenon, ensuring stability during streaming generation.

Specific kernel parameters, including stride, normalization, and selection modes, are detailed in the *Sparsity Config* section of Table 8 (Left).

## D. Efficient Deployment of Elastic Attention

A critical system bottleneck in hybrid attention pertains to the efficient scheduling of heterogeneous attention workloads within a single batch. Algorithm 1(a) illustrates the conventional paradigm, termed *Serial Dispatch*, as adopted by libraries like Flash-Attn[13] (Dao, 2024). This method entails explicit data rearrangement (highlighted in red), requiring the materialization of input tensors according to routing decisions $r$.

Consequently, it introduces two major sources of inefficiency that contradict the high-throughput requirements of long-context inference: (1) **Memory Overhead**, incurred by allocating and copying non-contiguous tensor fragments (e.g., separating retrieval heads from sparse heads); and (2) **Kernel Launch & Scheduling Overhead**. As noted in the main text, parallelism along the sequence dimension dominates execution in long-context scenarios. Launching separate kernels for different head groups fragments this workload, incurring latency from multiple kernel invocations and disrupting the GPU's ability to globally schedule thread blocks across available streaming multiprocessors.

To overcome these limitations, we employ the Block Sparse Attention (BSA) Kernel[14] (Guo et al., 2024) (Algorithm 1(b)). Obviating the need for tensor splitting, we pass routing decisions $r$ directly to the kernel as lightweight metadata $\mathbf{m}$. As depicted in the green block, the kernel leverages *thread-block level branching*: each thread block dynamically retrieves its assigned head's type from $\mathbf{m}$ and executes the corresponding attention logic. This design enables a **unified kernel launch** for all heads. By keeping the grid dimensions intact (Batch $\times$ Heads $\times$ Sequence Blocks), we effectively eliminate redundant memory copies and avoid workload fragmentation, allowing the GPU hardware scheduler to optimally distribute sequence blocks.

---

[13] https://github.com/Dao-AILab/flash-attention
[14] https://github.com/mit-han-lab/Block-Sparse-Attention

*Table 9.* LongBench-E results comparison. The 1st and the 2nd performance in each comparison group are highlighted with **bold font** and underlined, respectively.

| Method | Single-Document QA | | Multi-Document QA | | Summarization | | Few-shot Learning | | | Synthetic | | Code | | Avg. |
|---|---|---|---|---|---|---|---|---|---|---|---|---|---|---|
| | MF-en | Qasper | HotpotQA | 2WikiMQA | GovReport | MultiNews | TREC | TriviaQA | SAMSum | PCount | PRe | Lcc | RB-P | |
| **Qwen3-4B backbone model** | | | | | | | | | | | | | | |
| Qwen3-4B | 52.16 | 35.21 | 44.81 | 32.15 | 33.47 | 23.45 | 70.67 | 88.22 | 39.74 | 2.33 | 96.84 | 57.93 | 50.84 | 48.45 |
| + InfLLM-V2 | 49.13 | 37.51 | 40.26 | 29.68 | 33.99 | 25.92 | 67.67 | 86.14 | 38.76 | 4.30 | 72.78 | 62.56 | 56.59 | 46.68 |
| + DuoAttention | 48.85 | 34.60 | 42.25 | 29.18 | 33.26 | 23.68 | 66.33 | 87.70 | 39.73 | 2.11 | 92.24 | 57.57 | 50.25 | 46.95 |
| + PruLong | 48.88 | 34.28 | 45.14 | 30.01 | 33.36 | 23.70 | 66.33 | 88.31 | 39.77 | 4.79 | 89.67 | 55.44 | 50.95 | 47.19 |
| + MoBA | 42.41 | 35.80 | 38.76 | 29.84 | 33.82 | 25.11 | 68.67 | 85.69 | 39.34 | 2.33 | 72.52 | 58.61 | 50.63 | 45.09 |
| + NSA | 42.46 | 33.90 | 38.10 | 31.53 | 31.69 | 23.07 | 68.00 | 82.49 | 39.82 | 5.67 | 41.93 | 62.44 | 53.77 | 43.02 |
| + XAttention | 50.36 | 32.80 | 44.54 | 33.17 | 33.79 | 23.73 | 69.00 | 87.44 | 39.90 | 3.42 | 74.59 | 59.62 | 49.42 | 46.44 |
| + Elastic Attention (FA-SSA) | 49.13 | 35.27 | 47.87 | 29.85 | 33.35 | 23.65 | 69.33 | 87.23 | 40.62 | 2.00 | 94.86 | 57.80 | 50.87 | 48.08 |
| + Elastic Attention (FA-XA) | 52.06 | 36.74 | 45.65 | 33.18 | 33.45 | 23.55 | 68.67 | 87.33 | 39.79 | 4.39 | 84.32 | 50.47 | 58.11 | 47.59 |
| + Elastic Attention (XA-SSA) | 49.21 | 34.63 | 47.32 | 30.02 | 33.14 | 23.75 | 68.00 | 87.64 | 40.12 | 5.50 | 93.00 | 59.15 | 50.98 | 48.14 |
| **Qwen3-8B backbone model** | | | | | | | | | | | | | | |
| Qwen3-8B | 49.92 | 41.22 | 58.98 | 44.21 | 33.27 | 23.42 | 71.33 | 86.77 | 41.83 | 2.00 | 98.33 | 66.31 | 56.08 | 52.16 |
| + InfLLM-V2 | 46.35 | 38.05 | 49.51 | 35.14 | 34.49 | 24.67 | 67.00 | 87.03 | 39.61 | 12.07 | 79.67 | 66.86 | 52.30 | 49.03 |
| + DuoAttention | 49.59 | 41.32 | 51.82 | 37.22 | 33.14 | 23.19 | 70.33 | 86.83 | 42.11 | 0.00 | 95.00 | 67.43 | 57.32 | 50.67 |
| + PruLong | 51.06 | 41.04 | 53.87 | 39.90 | 33.20 | 23.39 | 70.33 | 87.68 | 41.83 | 0.33 | 97.00 | 67.30 | 57.18 | 51.34 |
| + MoBA | 48.81 | 40.65 | 48.26 | 38.29 | 35.19 | 25.47 | 67.67 | 84.13 | 40.85 | 14.00 | 83.89 | 68.21 | 56.89 | 50.47 |
| + NSA | 44.29 | 36.96 | 42.26 | 32.63 | 32.93 | 23.00 | 74.33 | 86.74 | 40.36 | 2.67 | 55.33 | 69.81 | 54.57 | 46.12 |
| + XAttention | 48.93 | 38.03 | 57.32 | 40.65 | 33.41 | 23.37 | 69.33 | 87.69 | 41.77 | 2.00 | 83.72 | 65.71 | 55.53 | 50.13 |
| + Elastic Attention (FA-SSA) | 51.45 | 40.84 | 53.34 | 39.74 | 33.14 | 23.24 | 72.33 | 88.67 | 41.56 | 0.00 | 96.14 | 68.44 | 57.47 | 51.51 |
| + Elastic Attention (FA-XA) | 48.18 | 39.83 | 59.38 | 40.60 | 33.30 | 23.29 | 69.00 | 87.78 | 41.92 | 2.33 | 99.56 | 65.40 | 55.74 | 51.66 |
| + Elastic Attention (XA-SSA) | 45.73 | 33.13 | 51.80 | 30.95 | 33.21 | 23.32 | 70.00 | 87.83 | 40.73 | 0.67 | 92.78 | 68.56 | 55.13 | 49.25 |
| **Llama-3.1-8B-Instruct backbone model** | | | | | | | | | | | | | | |
| Llama-3.1-8B-Instruct | 53.44 | 44.06 | 59.62 | 44.08 | 34.50 | 26.02 | 71.00 | 90.54 | 42.94 | 12.67 | 99.33 | 63.85 | 47.78 | 53.28 |
| + InfLLM-V2 | 48.60 | 38.93 | 50.74 | 41.85 | 34.40 | 25.76 | 69.00 | 89.92 | 43.04 | 6.33 | 77.73 | 59.78 | 68.81 | 50.73 |
| + NSA | 42.86 | 41.80 | 40.79 | 39.94 | 34.57 | 25.29 | 68.00 | 89.69 | 42.27 | 2.33 | 28.00 | 65.18 | 50.12 | 44.03 |
| + DuoAttention | 52.68 | 44.62 | 52.01 | 38.41 | 34.01 | 25.84 | 69.67 | 90.33 | 42.06 | 10.13 | 99.00 | 64.69 | 48.24 | 51.82 |
| + PruLong | 50.74 | 44.63 | 49.70 | 36.41 | 34.25 | 25.78 | 70.00 | 91.45 | 42.13 | 9.80 | 97.33 | 68.55 | 53.59 | 52.11 |
| + MoBA | 48.92 | 44.33 | 46.36 | 41.45 | 34.79 | 26.65 | 69.67 | 89.91 | 40.76 | 7.33 | 64.67 | 69.47 | 59.48 | 49.69 |
| + XAttention | 53.80 | 43.84 | 59.98 | 43.47 | 34.47 | 26.05 | 72.33 | 90.39 | 42.99 | 8.33 | 73.33 | 62.92 | 50.13 | 51.00 |
| + Elastic Attention (FA-SSA) | 53.48 | 46.35 | 55.50 | 42.33 | 34.50 | 25.77 | 69.67 | 91.70 | 42.60 | 9.33 | 98.67 | 67.72 | 53.69 | 53.35 |
| + Elastic Attention (FA-XA) | 53.65 | 45.14 | 60.21 | 45.67 | 34.56 | 26.03 | 71.67 | 91.37 | 42.62 | 8.31 | 91.00 | 63.22 | 49.75 | 52.71 |
| + Elastic Attention (XA-SSA) | 51.81 | 44.79 | 53.53 | 39.27 | 34.56 | 25.98 | 70.67 | 90.59 | 42.24 | 9.07 | 73.67 | 68.12 | 53.49 | 50.71 |

# E. Theoretical Explanation of Attention Router Optimization

The primary challenge in optimizing an attention router lies in the non-differentiable nature of discrete selection. To enable the model to learn which attention heads are essential for a given task, we employ a continuous relaxation technique based on the Gumbel-Softmax (specifically, the Gumbel-Sigmoid for binary decisions) and the Straight-Through Estimator (STE).

### E.1. Latent Representation and Logit Generation

Given the Key hidden states $x_K \in \mathbb{R}^{s \times H \times d'}$, where $s$ is the sequence length and $d$ is the hidden dimension, the router first extracts a task-aware representation via a pooling operation and a two-stage MLP:

$$x'_K = \text{Pooling}(x_K), x'_K \in \mathbb{R}^{H \times d'} \tag{10}$$

The routing logits $\mathbf{z}$ for each head are then computed as:

$$\mathbf{z} = \text{MLP}_{\text{router}}(\text{MLP}_{\text{task}}(x'_K)), \tag{11}$$

where $\mathbf{z} \in \mathbb{R}^H$ represents the unnormalized preference for a specific attention mode (e.g., FA vs. SA) for each head.

### E.2. Differentiable Sampling via Gumbel-Sigmoid

To simulate the stochasticity of discrete routing while maintaining differentiability, we apply the reparameterization trick (Bhaskar et al., 2025). We introduce i.i.d. noise samples $u \sim \text{Uniform}(0, 1)$ and transform them into Gumbel noise $g$:

$$g = -\log(-\log(u + \epsilon) + \epsilon), \tag{12}$$

where $\epsilon$ is a small constant for numerical stability.

The discrete binary decision is then relaxed into a continuous approximation $\hat{z}_{\text{soft}}$ using a temperature-dependent Sigmoid function:

*Table 10.* Performance on LongBench-E. We report average performance (Perf.) and $\Omega_{MSR}$ per task category. The 1st and the 2nd performance in each comparison group are highlighted with **bold font** and underlined, respectively. Note that $\Omega_{MSR}$ is not calculated for XA-SSA as it does not employ retrieval heads.

| Method | S-Doc QA | | M-Doc QA | | Summ | | In-Context | | Synthetic | | Code | | Avg. | |
|---|---|---|---|---|---|---|---|---|---|---|---|---|---|---|
| | Perf. | $\Omega_{MSR}$ | Perf. | $\Omega_{MSR}$ | Perf. | $\Omega_{MSR}$ | Perf. | $\Omega_{MSR}$ | Perf. | $\Omega_{MSR}$ | Perf. | $\Omega_{MSR}$ | Perf. | $\Omega_{MSR}$ |
| **Qwen3-4B backbone model** | | | | | | | | | | | | | | |
| Qwen3-4B | 43.69 | - | 38.48 | - | 28.46 | - | 66.21 | - | 49.59 | - | 54.38 | - | 48.45 | - |
| + MoBA | 38.16 | - | 34.30 | - | **29.46** | - | 64.57 | - | 37.42 | - | 54.62 | - | 45.09 | - |
| + NSA | 39.13 | - | 34.81 | - | 27.38 | - | 63.44 | - | 23.80 | - | **58.10** | - | 43.02 | - |
| + XAttention | 41.58 | - | 38.85 | - | 28.76 | - | 65.45 | - | 39.01 | - | 54.52 | - | 46.44 | - |
| + Elastic Attention (FA-SSA) | 42.20 | 0.66 | 38.86 | 0.68 | 28.50 | 0.76 | **65.73** | 0.73 | 48.43 | 0.71 | 54.34 | 0.82 | 48.08 | 0.73 |
| + Elastic Attention (FA-XA) | **44.40** | 0.68 | **39.42** | 0.71 | 28.49 | 0.82 | 65.26 | 0.76 | 44.35 | 0.74 | 54.29 | 0.87 | 47.59 | 0.76 |
| + Elastic Attention (XA-SSA) | 41.92 | - | 38.67 | - | 28.45 | - | 65.25 | - | **49.25** | - | 55.07 | - | **48.14** | - |
| **Qwen3-8B backbone model** | | | | | | | | | | | | | | |
| Qwen3-8B | 45.57 | - | 51.59 | - | 28.34 | - | 66.64 | - | 50.16 | - | 61.20 | - | 52.16 | - |
| + MoBA | 44.73 | - | 43.28 | - | **30.33** | - | 64.22 | - | 48.95 | - | 62.55 | - | 50.47 | - |
| + NSA | 40.63 | - | 37.45 | - | 27.97 | - | 67.14 | - | 29.00 | - | 62.19 | - | 46.12 | - |
| + XAttention | 43.48 | - | 48.99 | - | 28.39 | - | 66.26 | - | 42.86 | - | 60.62 | - | 50.13 | - |
| + Elastic Attention (FA-SSA) | **46.15** | 0.64 | 46.54 | 0.65 | 28.19 | 0.72 | **67.52** | 0.71 | 48.07 | 0.65 | **62.95** | 0.78 | 51.51 | 0.69 |
| + Elastic Attention (FA-XA) | 44.01 | 0.75 | **49.99** | 0.76 | 28.30 | 0.83 | 66.23 | 0.80 | **50.95** | 0.77 | 60.57 | 0.86 | **51.66** | 0.80 |
| + Elastic Attention (XA-SSA) | 39.43 | - | 41.38 | - | 28.27 | - | 66.19 | - | 46.73 | - | 61.85 | - | 49.25 | - |
| **Llama-3.1-8B-Instruct backbone model** | | | | | | | | | | | | | | |
| Llama-3.1-8B-Instruct | 48.75 | - | 51.85 | - | 30.26 | - | 68.16 | - | 56.00 | - | 55.81 | - | 53.28 | - |
| + MoBA | 46.63 | - | 43.91 | - | **30.72** | - | 66.78 | - | 36.00 | - | **64.48** | - | 49.69 | - |
| + NSA | 42.33 | - | 40.37 | - | 29.93 | - | 66.65 | - | 15.17 | - | 57.65 | - | 44.03 | - |
| + XAttention | 48.82 | - | 51.73 | - | 30.26 | - | **68.57** | - | 40.83 | - | 56.53 | - | 51.00 | - |
| + Elastic Attention (FA-SSA) | **49.92** | 0.64 | 48.92 | 0.63 | 30.17 | 0.73 | 67.99 | 0.74 | **54.00** | 0.64 | 60.70 | 0.79 | **53.35** | 0.69 |
| + Elastic Attention (FA-XA) | 49.40 | 0.71 | **52.94** | 0.69 | 30.30 | 0.80 | 68.55 | 0.79 | 49.66 | 0.72 | 56.49 | 0.87 | 52.71 | 0.77 |
| + Elastic Attention (XA-SSA) | 48.30 | - | 46.40 | - | 30.27 | - | 67.83 | - | 41.37 | - | 60.81 | - | 50.71 | - |

$$\hat{z}_{\text{soft}} = \sigma\left(\frac{\mathbf{z} + g}{\tau}\right) = \frac{1}{1 + \exp\left(-\frac{\mathbf{z}+g}{\tau}\right)} \tag{13}$$

Here, $\tau \in (0, \infty)$ is the temperature parameter. When $\tau \to \infty$, the distribution becomes uniform; as $\tau \to 0$, the output $\hat{z}$soft approaches a discrete Bernoulli distribution. In Section E.3, we introduce the strategy for parameter $\tau$.

### E.3. Temperature Annealing Schedule

The optimization process utilizes an annealing schedule for $\tau$ to bridge the gap between the continuous relaxation and the discrete reality. We define the decay as:

$$\tau^{(t)} = \max(\tau_{\min}, \tau_{\text{init}} \cdot \exp(-r \cdot p)), \tag{14}$$

where $p$ is the training step and $r$ is the decay rate (we set $r = 0.6$). In the early stages of training, a high $\tau$ encourages exploration by providing dense gradients; in later stages, a low $\tau$ forces the router to converge toward the "hard" binary decisions used during actual deployment.

## F. Additional Evaluation Results

In this section, we present detailed performance for the benchmarks discussed in the main paper.

### F.1. Evaluation Results on LongBench

Table 9 and Table 10 provides the complete evaluation results on the LongBench-E. We report the performance metrics across all sub-tasks, categorized into 6 categories: Single-Document QA (S-Doc QA), Multi-Document QA (M-Doc QA), Summarization (Summ), In-context Learning (In-Context), Synthetic, and Code Tasks.

*Table 11.* Detailed configuration for the RULER benchmark evaluation. We evaluate across exponentially increasing context windows up to 256k tokens.

| PARAMETER | CONFIGURATION DETAILS |
|---|---|
| **Context Windows** | 8k, 16k, 32k, 64k, 128k, 256k |
| **Sample Size** | 50 samples per task-length pair |
| **Evaluation Tasks** | **Retrieval (NIAH):** `single_{1-3}`, `multikey_{1-3}`, `multiquery`, `multivalue` 
 **QA & Extraction:** `qa_{1,2}`, `fwe` |

*Table 12.* Additional results on RULER and LongBench-v2.

| Models | RULER | | | | | | | | LongBench-v2 | | | |
|---|---|---|---|---|---|---|---|---|---|---|---|---|
| | 8K | 16K | 32K | 64K | 128K | 256K | Avg. Perf | $\Omega_{MSR}$ | Easy | Hard | Avg. Perf | Avg. $\Omega_{MSR}$ |
| *Qwen3-4B backbone model* | | | | | | | | | | | | |
| Qwen3-4B | 87.49 | 86.82 | 60.05 | 70.98 | 53.19 | 43.27 | 66.00 | - | 32.67 | 22.18 | 25.96 | - |
| + MoBA | 81.74 | 71.85 | 40.74 | 42.88 | 12.78 | 8.67 | 44.35 | - | 22.00 | **26.32** | 24.76 | |
| + NSA | **86.82** | 76.94 | 44.18 | 57.62 | 25.96 | 9.18 | 45.29 | - | 26.00 | 25.56 | 25.72 | |
| + XAttention | 85.93 | 84.60 | **60.32** | 67.01 | 48.76 | 38.42 | 61.23 | - | 26.00 | 24.81 | 25.24 | |
| + Elastic Attention (FA-SSA) | 83.35 | 79.24 | 50.79 | 67.03 | 47.83 | 47.32 | 61.81 | 0.66 | **34.00** | 24.44 | 27.88 | 0.70 |
| + Elastic Attention (FA-XA) | 86.56 | 85.38 | 56.88 | 69.42 | 49.48 | 43.47 | 63.27 | 0.67 | 32.00 | 25.94 | **28.12** | 0.72 |
| + Elastic Attention (XA-SSA) | 81.43 | 80.68 | 51.12 | 65.69 | 49.96 | 53.30 | 63.70 | - | 26.00 | 25.94 | 25.96 | - |
| *Qwen3-8B backbone model* | | | | | | | | | | | | |
| Qwen3-8B | 89.69 | 85.62 | 63.23 | 82.39 | 65.84 | 66.71 | 75.74 | - | 39.33 | 27.82 | 31.97 | - |
| + MoBA | 85.17 | 75.26 | 47.88 | 51.64 | 34.68 | 29.64 | 55.68 | - | 28.00 | 24.44 | 25.72 | |
| + NSA | 78.64 | 65.82 | 45.77 | 43.04 | 32.21 | 24.17 | 45.18 | - | 34.67 | 27.44 | 30.05 | |
| XAttention | 83.88 | 84.88 | 63.23 | 80.10 | 63.11 | 62.49 | 72.68 | - | 32.67 | 26.69 | 28.85 | |
| + Elastic Attention (FA-SSA) | **86.62** | 82.81 | 64.55 | 77.41 | 61.17 | 61.75 | 71.74 | 0.65 | 37.33 | **31.20** | **33.41** | 0.66 |
| + Elastic Attention (FA-XA) | 85.07 | **85.12** | **65.08** | **82.34** | **64.57** | **63.41** | **73.87** | 0.76 | 30.68 | 30.77 | 30.74 | 0.78 |
| + Elastic Attention (XA-SSA) | 71.48 | 74.49 | 56.35 | 69.03 | 64.56 | 55.73 | 66.31 | - | **38.00** | 29.70 | 32.69 | - |
| *Llama-3.1-8B-Instruct backbone model* | | | | | | | | | | | | |
| Llama-3.1-8B-Instruct | 92.88 | 92.83 | 89.46 | 70.79 | 80.12 | 72.34 | 83.47 | - | 32.00 | 33.08 | 32.69 | - |
| + MoBA | 89.05 | 67.14 | 30.12 | 6.13 | 1.15 | 0 | 37.38 | - | 10.00 | 12.78 | 11.78 | - |
| + NSA | 73.27 | 50.68 | 21.39 | 22.52 | 15.30 | 11.42 | 30.58 | - | 21.33 | 25.19 | 23.80 | |
| + XAttention | **93.11** | 89.74 | 86.11 | 66.89 | 75.55 | 41.6 | 75.36 | - | 26.67 | 28.2 | 27.64 | |
| + Elastic Attention (FA-SSA) | 89.93 | 83.42 | 80.20 | 56.30 | 68.16 | 56.47 | 72.85 | 0.65 | 28.00 | 29.70 | 29.09 | 0.73 |
| + Elastic Attention (FA-XA) | 92.82 | **92.00** | **87.80** | **68.23** | **78.87** | **68.51** | **81.82** | 0.72 | 30.67 | **30.83** | **30.77** | 0.75 |
| + Elastic Attention (XA-SSA) | 89.07 | 82.99 | 77.74 | 58.86 | 64.48 | 47.68 | 70.27 | - | **30.67** | 28.20 | 29.09 | - |

## F.2. Evaluation Results on LongBench-V2 and RULER

This subsection details the experimental settings and evaluation results for LongBench-V2 and the RULER benchmark. For RULER, as detailed in Table 11 and Table 13, our evaluation on the RULER benchmark covers a comprehensive range of context lengths, extending from 8K to 262K tokens. The comparative performance results are summarized in Table 12.

## G. Ablation Study

### G.1. Task-Discriminative Geometry of the Attention Router

To elucidate the operational logic of the Attention Router, we examine the geometric structure of the latent space learned by its internal projection layers. We hypothesize that, despite being optimized solely for the sparsity-performance trade-off without explicit task supervision, the router implicitly learns task-discriminative representations to facilitate optimal retrieval head allocation.

Formally, let $\boldsymbol{x}'_K \in \mathbb{R}^{H \times d'}$ denote the pooled hidden state of an input sequence. The $\text{MLP}_{\text{task}}$ maps this context to a latent routing representation $\mathcal{X}_K = \text{MLP}_{\text{task}}(\boldsymbol{x}'_{\mathbf{K}})$. To quantify the distinctness of these representations while mitigating the representation anisotropy inherent in pre-trained models, we employ a Pairwise Conditional Rescaling strategy. This approach normalizes the local subspace spanned by a specific task pair $(u, v)$ to measure their geometric relationship independent of global variances. We define the pairwise similarity metric $M_{uv}$ as follows. Let $\mathcal{X}_k$ be the set of latent

*Table 13.* Performance comparison of different attention methods on RULER subtasks. Tasks are grouped logically to highlight performance variations. Baseline corresponds to the respective foundation models' intrinsic attention mechanisms.

| Base Model | Method | NIAH Single | | | NIAH Multikey | | | NIAH Multi | | QA | | FWE | Avg. |
|---|---|---|---|---|---|---|---|---|---|---|---|---|---|
| | | 1 | 2 | 3 | 1 | 2 | 3 | Val | Qry | 1 | 2 | | |
| Qwen3-4B | Baseline | **100.00** | 80.00 | **95.02** | **73.05** | **47.39** | **37.54** | **65.91** | 66.19 | 40.14 | **35.59** | 84.47 | **66.00** |
| | Elastic Attn (FA-SSA) | 99.33 | **86.67** | 94.00 | 61.00 | 46.33 | 32.67 | 56.67 | 54.67 | 29.67 | 33.67 | **85.22** | 61.81 |
| | Elastic Attn (FA-XA) | 99.67 | 78.00 | 94.67 | 71.67 | 42.67 | 33.33 | 59.42 | 62.75 | **36.00** | 35.33 | 82.44 | 63.27 |
| | InfLLM-V2 | **100.00** | 60.00 | 75.20 | 46.40 | 31.20 | 19.60 | 50.40 | 53.90 | 22.80 | 26.80 | 74.93 | 51.02 |
| | DuoAttention | 99.00 | 86.00 | 94.33 | 56.67 | 24.33 | 16.00 | 59.67 | 62.08 | 28.67 | 31.67 | 82.89 | 58.30 |
| Llama3.1-8B | Baseline | **99.67** | 94.00 | **99.00** | **92.67** | **69.00** | **71.33** | **95.17** | **96.50** | **70.00** | 48.67 | 82.11 | **83.47** |
| | Elastic Attn (FA-SSA) | 96.33 | 91.33 | 97.33 | 62.00 | 59.33 | 63.67 | 80.58 | 85.33 | 41.00 | 39.67 | **84.67** | 72.85 |
| | Elastic Attn (FA-XA) | **99.67** | **94.33** | 98.67 | 90.33 | 67.00 | 67.33 | 92.42 | 95.08 | 63.67 | **49.67** | 81.89 | 81.82 |
| | InfLLM-V2 | 86.00 | 73.20 | 83.20 | 57.60 | 30.40 | 34.80 | 60.00 | 62.60 | 40.00 | 38.80 | 83.47 | 59.10 |
| | DuoAttention | 75.00 | 78.67 | 94.00 | 64.67 | 29.00 | 34.67 | 79.08 | 80.17 | 41.33 | 37.33 | 78.22 | 62.92 |

*Table 14.* Performance comparison on LongBench and RULER. The performance drop relative to the respective base model is shown in parentheses.

| Model | LongBench | RULER |
|---|---|---|
| **Qwen3-4B** | **48.45** | **66.00** |
| Elastic Attention (FA-SSA) | 48.08 (-0.37) | 61.81 (-4.19) |
| Elastic Attention (FA-XA) | 47.59 (-0.86) | 63.27 (-2.73) |
| LyChee (Lin et al., 2026) | 47.83 (-0.62) | 50.92 (-15.08) |
| **Qwen3-8B** | **52.16** | **75.74** |
| Elastic Attention (FA-SSA) | 51.51 (-0.65) | 71.74 (-4.00) |
| Elastic Attention (FA-XA) | 51.66 (-0.50) | 73.87 (-1.87) |
| LyChee (Lin et al., 2026) | 51.23 (-0.93) | 56.86 (-18.88) |

representations corresponding to task $k$. We first construct the local support set $\mathcal{S}_{uv} = \mathcal{X}_u \cup \mathcal{X}_v$ and compute the feature-wise standard deviation $\boldsymbol{\sigma}_{uv}$ within this subset:

$$\boldsymbol{\sigma}_{uv} = \sqrt{\frac{1}{|\mathcal{S}_{uv}|} \sum_{\mathbf{x} \in \mathcal{S}_{uv}} (\mathbf{x} \odot \mathbf{x}) + \epsilon}, \tag{15}$$

where $\odot$ denotes the element-wise product, and the calculation assumes zero-centered activations. Using these statistics, we define a projection $\phi_{uv}(\mathbf{x}) = \mathbf{x} \oslash \boldsymbol{\sigma}_{uv}$ (where $\oslash$ denotes element-wise division). The metric $M_{uv}$ is then calculated as the cosine similarity between the projected task centroids:

$$M_{uv} = \text{CosSim}\left(\frac{1}{|\mathcal{X}_u|} \sum_{\mathbf{x} \in \mathcal{X}_u} \phi_{uv}(\mathbf{x}), \ \frac{1}{|\mathcal{X}_v|} \sum_{\mathbf{x} \in \mathcal{X}_v} \phi_{uv}(\mathbf{x})\right). \tag{16}$$

As illustrated in Figure 9b, the latent space exhibits distinct modularity. The observation that similarity scores approach zero ($M_{uv} \approx 0$) implies that $\text{MLP}_{\text{task}}$ maps different problem types to orthogonal subspaces of the local manifold. This geometric orthogonality confirms that the router implicitly functions as a semantic discriminator, disentangling task representations into independent basis directions to apply specialized sparsity policies without requiring explicit task labels.

## G.2. Attention Mode Routing Analysis

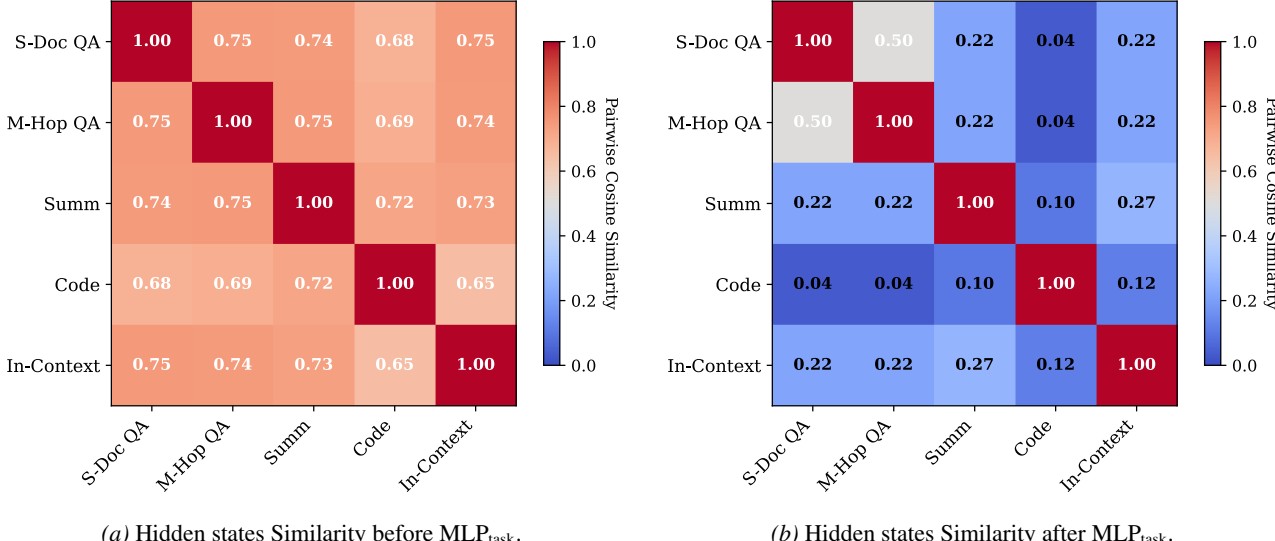

*(a)* Hidden states Similarity before MLP$_\text{task}$.

*(b)* Hidden states Similarity after MLP$_\text{task}$.

*Figure 9.* Pairwise cosine similarity of routing representations $\mathbf{z_{task}}$. The prevalence of near-zero scores ($M_{uv} \approx 0$) indicates that the router maps distinct tasks to orthogonal subspaces on the local manifold. This confirms that the model implicitly disentangles task semantics into independent directions without supervision.

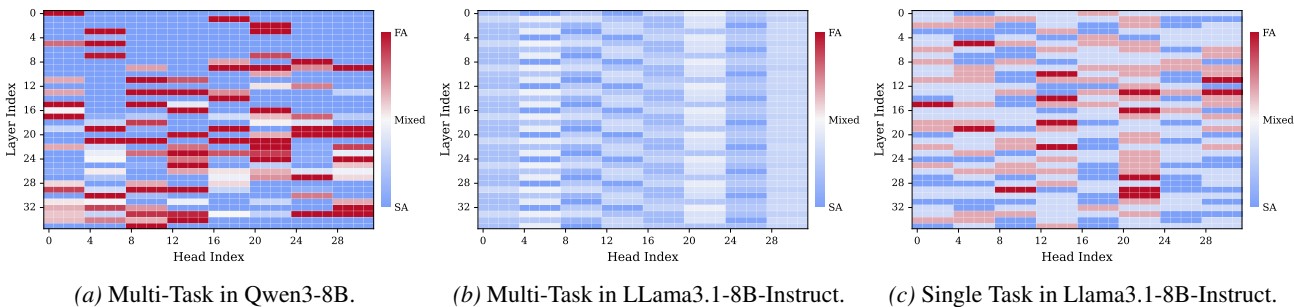

*(a)* Multi-Task in Qwen3-8B.

*(b)* Multi-Task in LLama3.1-8B-Instruct.

*(c)* Single Task in Llama3.1-8B-Instruct.

*Figure 11.* **Extended Head Robustness Analysis.** Similar to Figure 6, these heatmaps visualize the frequency of full-attention activation for each head. (a) and (b) show the multi-task global robustness for Qwen3-8B and Llama3.1-8B-Instruct, respectively, indicating how attention patterns generalize across tasks. (c) presents the robustness analysis for Llama3.1-8B-Instruct in a single-task setting. Darker blue indicates heads that are universally sparse, while darker red indicates heads that are universally active.

Comparing the multi-task heatmaps reveals a fundamental divergence in attention mechanisms between model families. For Qwen3 (Fig. 11a), we observe **structural universality**: a specific subset of heads remains consistently active (dark red) or sparse (dark blue) across all tasks, suggesting a task-agnostic attention topology. In contrast, Llama3.1 exhibits **context-dependent sparsity**. While the multi-task aggregate (Fig. 11b) shows no universally active heads, the single-task breakdown (Fig. 11c) confirms that strong activations exist but shift dynamically depending on the input context. This indicates that Qwen3 relies on fixed retrieval heads, whereas Llama3.1-8B-Instruct adaptively reallocates attentional resources based on task requirements.

We observe that during training, the effective sparsity levels gradually diverge across different tasks. Despite

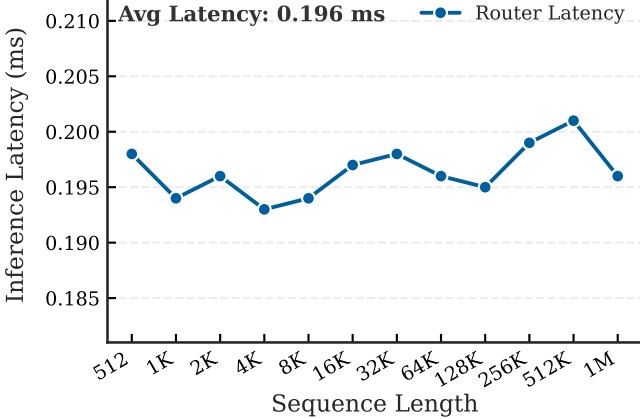

*Figure 10.* **Router latency analysis.** The router incurs negligible overhead (avg. 0.196 ms). Our design ensures length-invariant stability, maintaining constant speed from 512 to 1M tokens.

sharing the same non-tight constraint $t$, task-dependent differences emerge. This behavior is enabled by the Lagrangian

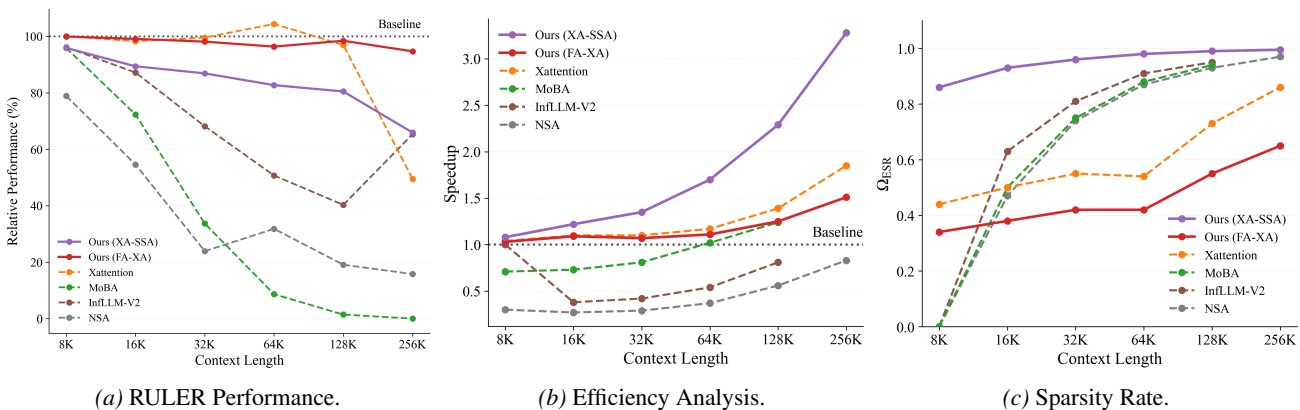

*(a)* RULER Performance.        *(b)* Efficiency Analysis.        *(c)* Sparsity Rate.

*Figure 12.* Analysis of length extrapolation capability and sparsity dynamics on the RULER benchmark (8K-256K). We adopt Llama-3.1-8B-Instruct as the backbone model to compare our Elastic Attention variants (FA-XA and XA-SSA) with including MoBA and NSA. (a) reports the RULER performance scores; (b) and (c) illustrate the trade-off between inference speedup and ($\Omega_{\mathrm{ESR}}$), highlighting the superior Pareto frontier established by our method.

constraint, which dynamically adjusts the weighting of task losses, allowing each task to tolerate different gaps between the achieved sparsity $\Omega_{\mathrm{MSR}}$ and the target $t$.

### G.3. Length Extrapolation and Sparsity Dynamics Analysis

We evaluate the length extrapolation capability on the RULER benchmark (8K-256K) using the Llama-3.1-8B-Instruct backbone. Figure 12 visualizes the interplay between performance, inference speedup, and $\Omega_{ESR}$. As the context length extends to 256K, baselines like MoBA and NSA suffer catastrophic degradation (near-zero accuracy). In contrast, our Elastic Attention (FA-XA) demonstrates superior robustness, maintaining a high score of 68.51. Even our highly efficient variant, Elastic Attention (XA-SSA), achieves a score of 47.68, significantly outperforming the standard Xattention baseline (35.82) while operating at a much higher sparsity.

Figure 12 (b) & (c) highlight a critical efficiency advantage. Baselines like NSA and InfLLM-V2 achieve high sparsity ($> 0.95$) but suffer from limited or regressed speedups ($< 1.0\times$) due to heavy dynamic selection overheads or kernel constraints. Conversely, our XA-SSA configuration achieves an extreme sparsity of $\sim 0.995$ while delivering a massive $3.28\times$ speedup, verifying the minimal overhead of our router. Meanwhile, Elastic Attention (FA-XA) strikes a balanced trade-off, securing $1.51\times$ acceleration while retaining essential information.

In summary, Elastic Attention establishes a superior Pareto frontier. Elastic Attention (FA-XA) prioritizes information retention to effectively mitigate "context collapse", while Elastic Attention (XA-SSA) maximizes throughput through adaptive extreme sparsity. Both configurations consistently outperform comparison methods in their respective regimes of accuracy and efficiency.

### G.4. Loss Curve and Monitoring metrics

To validate the training stability and the dynamic routing capabilities of our proposed **Elastic Attention**, we visualize the detailed training dynamics in Figure 13. This visualization decomposes the optimization process into four key perspectives: the primary language modeling loss, the sparsity regularization loss, the evolution of the routed sparsity metric ($\Omega_{\mathrm{MSR}}$), and the adaptive coefficients ($\lambda$).

**Optimization Stability.** As illustrated in Figure 13a and Figure 13b, the joint optimization of the language modeling objective and the Attention Router parameters is stable. The LM loss decreases rapidly and stabilizes around 2.1, confirming that the integration of the lightweight Attention Router and the injection of sparsity do not hinder the backbone model's convergence. Concurrently, the sparsity regularization loss drops significantly within the first 100 steps (from $\sim$0.16 to $\sim$0.06), indicating that the continuous relaxation scheme (Gumbel-Softmax) effectively guides the router to satisfy the prescribed sparsity constraints.

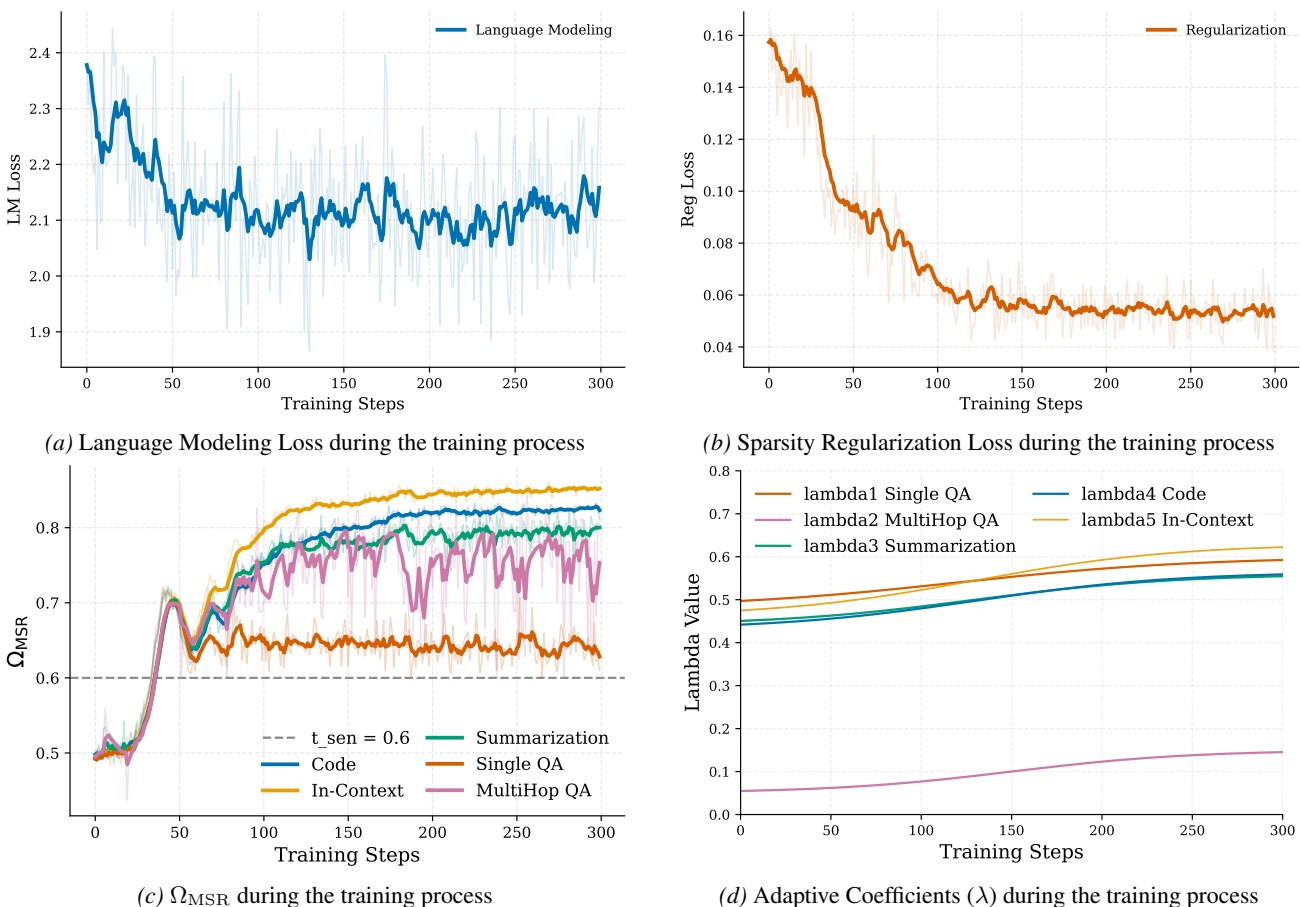

*(a)* Language Modeling Loss during the training process

*(b)* Sparsity Regularization Loss during the training process

*(c)* $\Omega_{\mathrm{MSR}}$ during the training process

*(d)* Adaptive Coefficients ($\lambda$) during the training process

*Figure 13.* Decomposition of Training Objectives for Elastic Attention. We visualize the training dynamics of the Attention Router, separating the total loss into (a) the primary language modeling objective and (b) the sparsity regularization term. Subfigures (c) and (d) illustrate the task-level differentiation in sparsity allocation ($\Omega_{\mathrm{MSR}}$) and adaptive coefficients ($\lambda$), demonstrating how the model automatically distinguishes between sparsity-robust and sparsity-sensitive tasks.

**Differentiation in Elastic Attention Allocation ($\Omega_{\mathrm{MSR}}$).** Figure 13c provides empirical evidence for the motivation described in Section 1: downstream tasks naturally exhibit different sensitivity to attention sparsity.

- **Task-Dependent Routing:** Starting from a neutral initialization, the Attention Router automatically learns to differentiate between tasks. Consistent with our hypothesis, **sparsity-sensitive tasks** (such as Code and In-Context learning) converge to higher $\Omega_{\mathrm{MSR}}$ values (approx. 0.80–0.85), indicating a higher allocation of Full Attention (FA) computation to preserve performance.

- **Sparsity-Robust Efficiency:** Conversely, **sparsity-robust tasks** like Q&A task plateau at lower values, closer to the target threshold (dashed line, representing $t_{\mathrm{sen}}$). This confirms that Elastic Attention successfully identifies tasks that can tolerate higher sparsity levels, thereby improving inference throughput without unnecessary computation.

**Adaptive Coefficients ($\lambda$).** Finally, Figure 13d plots the evolution of the Lagrangian multipliers ($\lambda$), which dynamically scale the penalty for violating sparsity targets. We observe that $\lambda_5$ (In-Context) increases most aggressively, suggesting that the model prioritizes fulfilling the density requirements for this sensitive task. This adaptive mechanism ensures that the trade-off between computational cost and model quality is balanced automatically, removing the need for the manual task-specific tuning.

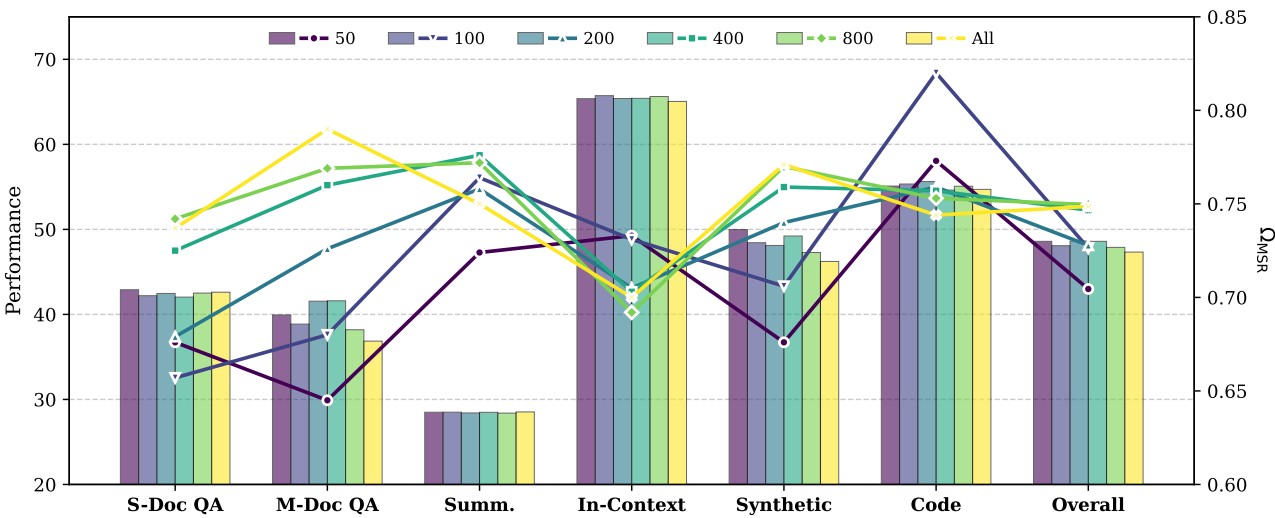

*Figure 14.* Impact of router input truncation length on downstream performance and $\Omega_{\text{MSR}}$. We compare varying truncation budgets ($L \in \{50, \ldots, 800, \text{All}\}$) applied to the concatenation of the sequence's prefix and suffix. Results indicate that increasing the input length beyond 100 tokens yields negligible performance gains and may degrade router selectivity due to a lower signal-to-noise ratio.

## G.5. Impact of Input Truncation on Task Identification

To optimize the trade-off between routing efficiency and accuracy, we investigate the sensitivity of the Attention Router to the input sequence length. Specifically, we analyze how varying the truncation budget influences the router's ability to distinguish between task types and allocate appropriate sparsity patterns. Figure 14 illustrates the performance and sparsity trends across six downstream tasks as the truncation budget varies from 50 tokens (pooling boundaries) to the full sequence.

**The Boundary-Pooling Hypothesis.** Our default strategy employs a boundary-pooling mechanism that aggregates only the first and last 100 tokens. This design is predicated on the observation that task-specific instructions (System Prompts) typically reside at the beginning of the context, while specific user queries appear at the end. The intermediate content often consists of long context (e.g., documents to be summarized) which, while necessary for the *generation* phase, acts as noise for the *routing* phase.

**Analysis of Signal-to-Noise Ratio.** As shown in Figure 14, we observe that performance generally saturates around a truncation length of 100-200 tokens. Notably, incorporating the full sequence does not lead to performance improvements and, in some cases (e.g., Multi-Document QA), results in suboptimal routing decisions. We attribute this to a dilution of the task identification signal: as the router processes more tokens from the document body, the distinct semantic signatures of the instructions become obscured by the high variance of the content. Consequently, the router struggles to classify the task type accurately, leading to a convergence in sparsity patterns (as seen in the flattened sparsity lines for longer lengths) without a corresponding gain in generation quality. These findings validate our selection of a 100-token boundary window as a robust configuration that captures essential task descriptors while filtering out content-induced noise.

# H. Error Analysis

In Table 15, 16, and 17, we present representative model outputs comparing our method with other baselines. Due to the extensive length of the contexts, only a partial input context is shown. We observe that the primary source of performance improvement stems from our method's ability to accurately identify and respond to the key contextual segments relevant to the query.

---

**Case 1: Policy Decision Making (Long-Context)**

*Context:*

"The shaded areas of the map indicate ESCAP members and associate members.* The Economic and Social Commission for Asia and the Pacific (ESCAP) is the most inclusive intergovernmental platform in the Asia-Pacific region. The Commission promotes cooperation among its 53 member States and 9 associate members in pursuit of solutions to sustainable development challenges. ESCAP is one of the five regional commissions of the United Nations. The ESCAP secretariat supports inclusive, resilient, and sustainable development in the region by generating action-oriented knowledge, and by providing technical assistance and capacity-building services in support of national development objectives, regional agreements, and the implementation of the 2030 Agenda for Sustainable Development. *The designations employed and the presentation of material on this map do not imply the expression of any opinion whatsoever on the part of the Secretariat of the United Nations con..."

---

*Question:*

Which policy mix should the government pursue to best balance fiscal sustainability, private sector engagement, and the energy transition, while maintaining political and social stability?

---

**Correct Prediction (Ours / Ground Truth):** ✓

**Gradual Energy Transition with National Green Investment Bank:** Create a national green investment bank... **Maintain existing fossil fuel subsidies for the next five years** to ensure energy price stability while gradually scaling up renewable energy... **while postponing the introduction of a carbon tax**.

---

**Incorrect Baselines:**

**Qwen3-8B, PruLong, DuoAttn, InfLLM-V2**:
**Aggressive Fiscal Realignment...:** Introduce a substantial **carbon tax** on oil production... **Gradually phase out fossil fuel subsidies** over the next five years... *(Error: Premature carbon tax and subsidy removal contradicts stability goals)*

**MoBA**:
**Immediate Fossil Fuel Subsidy Removal...: Remove all fossil fuel subsidies immediately**... Introduce a carbon pricing mechanism within two years... *(Error: "Immediate removal" ignores social stability constraint)*

**NSA**:
**Blended Finance and Export-Led...:** Establish a public-private blended finance fund... **Implement a modest carbon tax**... *(Error: Incorrect focus on export markets and carbon tax timing)*

---

*Figure 15.* Qualitative comparison on a complex policy reasoning task. Our model correctly identifies the 'Gradual' approach required for stability, whereas baselines hallucinate 'Aggressive' or 'Immediate' measures that contradict the stability constraint.

---

**Case 2: Legal Reasoning & Document Understanding**

*Context:*
Current to June 20, 2024 Last amended on June 20, 2024 À jour au 20 juin 2024 Dernière modification le 20 juin 2024 Published by the Minister of Justice at the following address: http://laws-lois.justice.gc.ca Publié par le ministre de la Justice à l'adresse suivante : http://lois-laws.justice.gc.ca CANADA CONSOLIDATION Financial Administration Act CODIFICATION Loi sur la gestion des finances publiques R.S.C., 1985, c. F-11 L.R.C. (1985), ch. F-11 Current to June 20, 2024 Last amended on June 20, 2024 À jour au 20 juin 2024 Dernière modification le 20 juin 2024 OFFICIAL STATUS OF CONSOLIDATIONS CARACTÈRE OFFICIEL DES CODIFICATIONS Subsections 31(1) and (2) of the Legislation Revision and Consolidation Act, in force on June 1, 2009, provide as follows: Les paragraphes 31(1) et (2) de la Loi sur la révision et la codification des textes législatifs, en vigueur le 1er juin 2009, prévoient ce qui suit : Published consolidation is evidence Codifications comme élément de preuve 31 (1) Ever...

---

*Question:*
The "Special Economic Measures Act (SEMA)" and the "Financial Administration Act (FAA)" have distinct roles... **how might their application differ in a hypothetical scenario where a Canadian citizen's business assets are seized...?** Which interpretation best reflects the nuanced interplay?

---

**Correct Prediction (Ours / Ground Truth):**                                                                     ✓
SEMA allows the government to freeze the assets as part of a broader sanctions framework without immediate concern for forfeiture, whereas the FAA would require a legal process to determine whether the assets should be **permanently seized and reallocated for public use**.

---

**Incorrect Baselines:**

**Qwen3-8B, PruLong, DuoAttn, MoBA, NSA**:
SEMA allows for freezing... while the FAA would require an **assessment of the legal grounds** for the potential forfeiture... ensuring **compliance with Canadian law**... *(Error: Generic response. Fails to identify the specific outcome of "public use" reallocation.)*

**InfLLM-V2**:
The FAA would necessitate a legal review to determine whether the assets should be seized for governmental use, while SEMA would enable freezing... **delaying any immediate decision**... *(Error: Incorrectly focuses on procedural delays rather than the substantive distinction in asset destination.)*

*Figure 16.* Comparison on a bilingual legal document. Our model accurately extracts the specific legal provision regarding asset **reallocation for public use** (FAA), whereas baselines provide generic descriptions of "legal assessments" or "compliance" without specific details.

---

**Case 3: Long-Context Narrative Understanding (The Count of Monte Cristo)**

*Context:*

Chapter 1: Marseilles – The Arrival. On the 24th of February, 1810, the look-out at Notre-Dame de la Garde signalled the three-master, the Pharaon from Smyrna, Trieste, and Naples. As usual, a pilot put off immediately, and rounding the Chateau d'If, got on board the vessel between Cape Morgion and Rion island. Immediately, and according to custom, the ramparts of Fort Saint-Jean were covered with spectators; it is always an event at Marseilles for a ship to come into port, especially when this ship, like the Pharaon, has been built, rigged, and laden at the old Phocee docks, and belongs to an owner of the city. The ship drew on and had safely passed the strait, which some volcanic shock has made between the Calasareigne and Jaros islands; had doubled Pomegue, and approached the harbor under topsails, jib, and spanker, but so slowly and sedately that the idlers, with that instinct which is the forerunner of evil, asked one another what misfortune could have happene...

---

*Question:*

Who were the ones who framed Edmond Dantes by writing and sending a letter filled with lies about him to the public prosecutor?

---

**Correct Prediction (Ours / Ground Truth):** ✓

**Danglars, Fernand and Caderousse**.

---

**Incorrect Baselines:**

**Llama3.1-8B-Instruct, DuoAttn, NSA**:
**Fernand, Villefort and Danglars**
*(Error: Entity Confusion - Mistook the prosecutor Villefort for a conspirator)*

**InfLLM-V2**:
**Fernand and Caderousse**
*(Error: Incomplete - Failed to retrieve the mastermind Danglars)*

**PruLong**:
**Caderousse, Villefort and Fernand**
*(Error: Entity Confusion - Incorrectly included Villefort)*

*Figure 17.* **Qualitative comparison on narrative entity tracking.** The task requires identifying the specific characters who conspired to frame the protagonist. Our model accurately retrieves the correct trio, whereas baselines consistently hallucinate "Villefort" (the public prosecutor) into the group, failing to distinguish between the plotters and the judicial figure involved later.

