# OpenReview forum: "Elastic Attention: Test-time Adaptive Sparsity Ratios for Efficient Transformers"
_ICML.cc/2026/Conference — ICML 2026 regular_

### Official Review · Reviewer_EANn · 2026-03-01

**Soundness:** 3
**Presentation:** 3
**Significance:** 3
**Originality:** 3
**Overall Recommendation:** 4
**Confidence:** 5

**Summary:**

This work proposes mixing full attention and sparse attention at the head level, i.e., elastic attention, to accelerate long-context inference. Elastic attention features a lightweight router selecting whether to run full or sparse attention, and a fused kernel executes both efficiently. With the backbone frozen, the router is trained via a Gumbel/STE objective. Experiments on long-context benchmarks show a better accuracy–efficiency trade-off than static sparsity ratios.

**Compliance With Llm Reviewing Policy:**

Affirmed.

**Final Justification:**

I maintain my positive score.

**Key Questions For Authors:**

1. How should target sparsity constraints be set without task labels? The paper uses task-dependent constraints; can the authors provide a principled, label-free way to set/learn these intervals (or adapt them online) for new domains and unseen task mixtures?

2. The paper focuses on attention compute and kernel efficiency; can the authors report end-to-end wall-clock improvements (including routing, framework overhead, memory, and decode) in a standard serving stack, not just kernel-level speedup?

3. What is each specific loss term’s contribution?  Could authors provide objective ablations and resulting accuracy–sparsity curves, if possible.

4. Results suggest model-dependent behavior (e.g., different backbones benefiting from different FA–SA pairings). What properties of a model determine whether FA–SSA vs. FA–XA is preferable, and can this be predicted without exhaustive tuning?

**Limitations:**

yes

**Strengths And Weaknesses:**

Strengths

1. The motivation is well-grounded: the paper explicitly connects task-level behavior to sparsity allocation, and the proposed solution directly targets the mismatch introduced by static sparsity ratios.

2. While routing/gating resembles conditional computation ideas, applying it to attention mode selection at head granularity, together with a fused-kernel implementation tailored for long-context inference, is a meaningful and well-engineered contribution.

Weaknesses

1. The method still relies on **manually chosen target sparsity** regimes/intervals (e.g., “sparsity-sensitive” vs. “sparsity-robust”), and it is unclear how robust these choices are under distribution shift or for unseen task mixtures. The “two-regime” abstraction may be overly coarse for many real prompts that combine retrieval, synthesis, and reasoning in a single input.

2. Some deployment details could be made more explicit: when exactly is routing computed (once per prefill vs. updated across turns), how routing interacts with caching in multi-turn chat, and whether routing decisions are stable across slight prompt perturbations.

3. The conceptual novelty (dynamic sparsity allocation) is clear, but the paper could sharpen comparisons to the closest “dynamic sparse attention” or “learned sparsification” baselines beyond static hybrid-head methods, to better isolate what is uniquely enabled by the router + fused-kernel pairing.

---

> ### Author Rebuttal · Authors · 2026-03-29
>
> Thanks for your review. Due to the rebuttal word limit, we provide the anonymous URL (https://anonymous.4open.science/r/Elastic-Attention-D370/rebuttal/rebuttal.md) containing complete figures and tables.
>
> We warmly invite you to consult these materials following the guidance in our rebuttal.
>
> ---
>
> > **W1&Q1**:The "two-regime" abstraction seems overly coarse for mixed prompts and relies on manually chosen sparsity bounds.
>
> **R**: Thank you for this excellent question. We have addressed this question in our response to **Reviewer 6by5 (W2 & Q2)**. Please refer to that reply for detailed explanations!
>
> ---
>
> > **W2**: Lack of partial deployment and router work details.
>
> **R**: Thank you for raising this point. We believe some relevant details may have been overlooked in the paper, and we clarify them as follows:
>
> **(1) Inference efficiency**: The router introduces negligible overhead—its inference latency is only 0.196 ms (Figure 10), demonstrating that it is lightweight and practical for deployment.
>
> **(2) Applicability to multi-turn settings**: In multi-turn chat scenarios, the problem naturally reduces to long-form generation, similar to summarization tasks. When conversations become very long, one can re-trigger the prefill stage to allow the router to re-adapt and dynamically adjust sparsity according to the updated context.
>
> **(3) Robustness to input variation**: We have explicitly evaluated the impact of input perturbations (i.e., different samples within the same task) in Figure 11(c). The results show that the router learns fine-grained, underlying sparsity patterns rather than memorizing coarse task-level representations, ensuring both stability and adaptability across varying inputs.
>
> ---
>
> > **W3**: Comparison to the closest “dynamic sparse attention” or “learned sparsification” baselines beyond static hybrid-head methods.
>
> **R**: We believe there is a misunderstanding. PruLong, InfLLM-V2 (Tables 1 and 2), NSA, and MoBA (Tables 8 and 10) all fall under the category of current dynamic sparse attention methods and represent strong, competitive baselines.
>
> To further demonstrate the novelty of our Router combined with the fused kernel, we include a comparison with *LycheeDecode (ICLR 2026 Poster)*[1]. **The full comparison table is available at our anonymous link (rebuttal.md -> Reviewer EANn -> Weakness3)**. Elastic Attention can still demonstrates superior efficiency and performance compared to other methods.
>
> ---
>
> > **Q2**: Share the end-to-end wall-clock improvements (including routing, framework overhead, memory, and decode) in a standard serving stack.
>
> **R**: Thank you for this excellent question. However, we regret that we currently do not have access to a standard serving stack environment. Instead, we provide the requested end-to-end evaluation metrics, measured via inference over **500 mixed real-world task samples** from LongBench V1 to simulate realistic serving conditions. **We show the results at our anonymous link (rebuttal.md -> Reviewer EANn -> Question2)**.
>
> ---
>
> > **Q3**: Details of each specific loss term’s contribution.
>
> **R:** Thank you for this constructive question. We conducted ablation studies on the individual terms in $L_{\text{diff}}$ and the coefficient $\lambda$ (Eq. 8).
> **Detailed accuracy–sparsity curves and the full ablation table are available at our anonymous link (rebuttal.md -> Reviewer EANn -> Question3)**. Our results show that $L_{\text{diff}}$ is the primary driver of task-aware routing. Removing it causes the router to lose its ability to differentiate between tasks, leading to a severe -8.83% degradation. In contrast, $\lambda$ mainly serves to stabilize the training process, with a relatively minor impact on overall performance.
>
> ---
>
> > **Q4**: Can the preferred sparse pattern (SSA vs. XA) be predicted without exhaustive tuning?
>
> **R**: Thank you for raising this good question! While both SSA and XA modes demonstrate strong performance, their relative advantages can be unpredictable. We believe that their effectiveness is modular and independent of specific backbone architectures. In most cases, **we recommend using SSA, as it has proven to be effective while also providing superior acceleration benefits**.
>
> ---
>
> > **Reference**
>
> [1] Lin, Gang, et al. "LycheeDecode: Accelerating Long-Context LLM Inference via Hybrid-Head Sparse Decoding." arXiv preprint arXiv:2602.04541 (2026).
>
> ---
>
> We appreciate your review once again and hope our responses adequately address your concerns.
>
> If any further questions remain, please do not hesitate to raise them.

---

> > ### Author Rebuttal · Reviewer_EANn · 2026-04-03
> >
> > I maintain my positive score.

---

> > > ### Author Response · Authors · 2026-04-03
> > >
> > > Thank you for your reply and for supporting our work! We are confident that integrating your valuable insights will help us improve the manuscript!

---

### Official Review · Reviewer_6by5 · 2026-03-11

**Soundness:** 3
**Presentation:** 3
**Significance:** 3
**Originality:** 3
**Overall Recommendation:** 4
**Confidence:** 3

**Summary:**

Briefly summarize the paper and its contributions. This is not the place to critique the paper; the authors should generally agree with a well-written summary. This is also not the place to paste the abstract; please provide the summary in your own understanding after reading.

The paper proposes Elastic Attention, a test-time adaptive hybrid attention mechanism that dynamically routes attention heads between Full Attention  and Sparse Attention based on the input. A lightweight Attention Router makes per-head binary routing decisions in prefill stage. A fused kernel executes FA and SA heads jointly to avoid sequential overhead. Across Qwen3-(4B/8B) and Llama 3.1 8B backbones, Elastic Attention improves the performance–efficiency trade-off compared to static hybrid-head baselines (e.g., DuoAttention, PruLong) and training-free methods (e.g., XAttention), and adapts the model sparsity ratio Ω_MSR to task regimes without changing backbone parameters.

**Compliance With Llm Reviewing Policy:**

Affirmed.

**Final Justification:**

I will maintain the review score.

**Key Questions For Authors:**

1.	The router pools only boundary tokens (first/last 100), why not the full sequence? And what about the long-input scenario, will current pooling method cause error?
2.	The task-dependent sparsity constraint (lower/upper bounds) is under-specified. How are bounds chosen?
3.	The attention sparsity ratio is only dependent on the input. The decoding stage may differ in attention sparsity though they are of same task type. What is the relevance between two stages?
4.	In Figure3, can you explain the relationship between Head n and  retrieval/sparse head ?

**Limitations:**

Yes

**Strengths And Weaknesses:**

Strengths:
1.	The author evaluates the relationship between attention sparsity ratio and task type. Introduces a per-head, input-conditioned router that selects FA vs SA at test time, moving beyond the fixed ratios of prior hybrid-head designs.
2.	Appropriately coped with the problem appeared in training and inference, such as using Gumbel-Softmax + STE and fused kernel.
3.	Extensive experiments. Evaluates across three strong backbones and multiple long-context benchmarks. Includes ablations on router architecture (task MLP effect, hidden sizes), routing patterns across heads, and target sparsity hyperparameters.

Weekness：
1.	The router pools only boundary tokens (first/last 100); this could misclassify tasks where discriminative cues lie in the middle of the context. And what about the long-input scenario?
2.	In Section 3.2, the task-dependent sparsity constraint (lower/upper bounds) is under-specified.
3.	The attention sparsity ratio is only dependent on the input, i.e., the router only functions in prefill stage. The decoding stage may differ in attention sparsity though they are of same task type. The relevance between two stages isn’t clear.

---

> ### Author Rebuttal · Authors · 2026-03-29
>
> Thank you for your review. Below is our response, which aims to clarify potential misunderstandings and provide additional details.
>
> ---
>
> > **W1 & Q1**：Question about the boundary of router pools.
>
> **R**:  Thank you for raising this important point! We have indeed investigated this design choice extensively. Please refer to **Appendix H.5** for detailed analysis. Briefly, given that user queries typically concentrate **at the beginning or end of inputs**, pooling these boundaries suffices. We note that pooling the entire sequence remains robust (Figure 14), representing a deliberate **efficiency-effectiveness trade-off** that favors practical deployment without compromising performance.
>
> ---
>
> > **W2 & Q2**: Question about the task-dependent sparsity constraint (lower/upper bounds).
>
> **R**: Thank you for this astute observation. We respectfully suggest that Section 5.2 and Appendix H.4 provide detailed discussions on the sparsity bound *t* that may have been overlooked. In short, the selection of *t* requires minimal tuning because it functions as a soft, non-tight bound rather than a rigid target. In practice, one only needs to categorize the task as information-dense or information-sparse, and assign *t* accordingly (follow our paper's setting is ok). The model does not actually converge to these extremes during training; instead, it gradually approaches the appropriate sparsity level from both sides (see Figure 13c).
>
> For new tasks, our five training categories (Section 4.1) already cover the vast majority of scenarios. For entirely OOD tasks, we recommend following the probe experiment described in Section 2.1 to determine whether the task is information-dense or sparse. We find the results insensitive to this binary choice once the task nature is correctly identified through the preliminary probe.
>
> ---
>
> > **W3 & Q3**: Question about attention sparsity in the decoding stage and the relevance between the prefill and decode stages.
>
> **R**: Thank you for raising this good question! Decoding acceleration is indeed critical for long sequences. We first clarify the distinct roles (relevance) of these two stages: (1) prefill enables context comprehension, while decoding generates subsequent tokens conditioned on that context; (2) existing sparse attention research **primarily targets prefill acceleration for long inputs**, whereas **decoding efficiency can be achieved through orthogonal techniques** such as speculative decoding or Multi-Token Prediction (MTP); (3) furthermore, the dynamic nature of sparsity patterns during decoding presents significant **engineering challenges** for current kernel implementations.
>
> We respectfully note that our scope is explicitly limited to the prefill stage, as clearly stated in our Introduction: *"Elastic Attention, which enables the model to automatically adjust its overall sparsity during the prefill stage."*
>
> ---
>
> > **Q4**: Explanation about the relationship between Head n and retrieval/sparse head in Figure 3.
>
> **R**: We appreciate this question. We suspect there may be a minor confusion: Figure 3 actually depicts *"Head H"* rather than *"Head n"*.
> Please refer to Equation 3 first: each attention head *H* can operate as either a Retrieval Head or a Sparse Head, depending on the router assignment. In Figure 3, we distinguish these using **solid lines for Retrieval Heads** and **dashed lines for Sparse Heads** to illustrate how individual heads adopt different head patterns.
>
> ---
>
> We appreciate your review once again and hope our responses adequately address your concerns.
>
> If any further questions remain, please do not hesitate to raise them.

---

> > ### Author Rebuttal · Reviewer_6by5 · 2026-04-03
> >
> > - My concerns have been resolved and I am happy to keep my score.

---

> > > ### Author Response · Authors · 2026-04-03
> > >
> > > Thank you for your reply and for supporting our work! We are confident that integrating your valuable insights will help us improve the manuscript!

---

### Official Review · Reviewer_Cxqe · 2026-03-13

**Soundness:** 3
**Presentation:** 3
**Significance:** 3
**Originality:** 2
**Overall Recommendation:** 5
**Confidence:** 4

**Summary:**

The paper addresses the computational bottleneck of processing long contexts in LLMs. While hybrid attention models successfully mix Full Attention and Sparse Attention to improve efficiency, they rely on fixed, static ratios that cannot adapt to the varying complexity of different downstream tasks. To solve this, the authors introduce Elastic Attention, featuring a lightweight, trainable "Attention Router" that dynamically assigns each attention head to either Full Attention or Sparse Attention mode during inference based on the specific input sequence. This allows the model to automatically apply high sparsity to robust tasks (like summarization) and lower sparsity to sensitive tasks (like question answering), achieving state-of-the-art performance and inference speedups without requiring modifications to the pretrained backbone.

**Compliance With Llm Reviewing Policy:**

Affirmed.

**Key Questions For Authors:**

please refer to the weakness section above.

**Limitations:**

yes (but for limitation, they did not explicitly discuss)

**Strengths And Weaknesses:**

**Strengths**

* The proposed approach is intuitive. The effectiveness of sparse attention and retrieval heads has already been demonstrated, and connecting the two is a practical and worthwhile attempt.
* The introduction of a fused kernel enables an efficient implementation of sparse attention.
* The task similarity analysis and head-level analysis are conducted in great detail, helping readers understand why the proposed method works and how it operates in practice.
* The paper also provides error analysis and training process analysis, which help in understanding the experiments more thoroughly.

**Weaknesses**

* The method is evaluated on LongBench, RULER, and LongBench-V2, which mainly consist of short retrieval or multiple-choice QA tasks. I am curious how the method would perform on benchmarks that better reflect real LLM use cases, particularly those measuring generation quality. Tasks such as math reasoning or agentic coding could be considered. Personally, LongBench v1 and v2 seem quite similar, so including both may not be necessary.
* The method requires additional data and training. An analysis of whether this approach is scalable would be a meaningful direction for future work. It would also be important to examine whether the proposed routing method can be applied efficiently and maintain stable load balancing in a multi-node environment.
* Is there a reason why Figure 8(a) is not presented in the same format as (b) and (c)? The bar graph in (a) is difficult to read.

---

> ### Author Rebuttal · Authors · 2026-03-29
>
> Thanks for your review. Due to the rebuttal word limit, we provide the anonymous URL (https://anonymous.4open.science/r/Elastic-Attention-D370/rebuttal/rebuttal.md) containing complete figures and tables.
>
> We warmly invite you to consult these materials following the guidance in our rebuttal.
>
> ---
>
> > **W1**: Lack of results on real-world tasks.
>
> **R**: Thank you for this suggestion. We would like to clarify that LongBench v1 and v2 target real-world model capabilities from different perspectives: v1 emphasizes broad task coverage, while v2 focuses more on ultra-long, multi-hop reasoning, making them fundamentally different benchmarks.
>
> Following the reviewer’s advice, we have also supplemented additional evaluations on real-world generation tasks, including **mathematical reasoning** and **domain-specific long-context understanding** (LongHealth[1]). Results are presented in the table below.
>
> | Model | AIME24 | GSM8K | Math | LongHealth | AVG |
> |:---:|:---:|:---:|:---:|:---:|:---:|
> | Qwen3-4B | 6.70 | 43.10 | 55.80 | 63.92 | 42.38 |
> | DuoAttention | 6.70 | **45.80** | 52.40 | 49.98 | 38.72 (-3.66) |
> | Elastic Attention (FA-SSA) | **10.00** | **45.80** | **57.10** | 59.42 | 43.08 (+0.70) |
> | Elastic Attention (FA-XA) | 7.10 | 45.60 | 56.30 | **64.4** | **43.35** (+0.97) |
>
> ---
>
> > **W2**: Require discussion about the scalability of our method, and how our method can be employed in a multi-node environment.
>
> **R**: This is a good question! These two concerns indeed address the same practical consideration. We respectfully ask the reviewer to refer to our **Impact Statement (Lines 447–460)**, where we discuss deployment scenarios and acknowledged limitations. Technically, the dynamic head-wise allocation of sparse and full heads inherently prevents balanced load distribution across multiple nodes (Lines 452–453). Consequently, our method is specifically designed for on-device deployment of small-to-medium models (Lines 453–454), where balancing latency and performance is paramount. Under this design philosophy, scaling across multiple nodes offers limited practical benefit, as we intentionally prioritize single-node efficiency for edge-computing scenarios.
>
> ---
>
> > **W3**: Figure 8(a) has inconsistent formatting with (b)/(c) and the bar graph is hard to read.
>
> **R**: Thank you for pointing out this formatting issue. We apologize for the reading inconvenience, and have redrawn Figure 8(a) to fully match the format of (b) and (c) for clear cross-comparison as suggested. **The revised figure is available in the anonymous link (rebuttal.md -> Reviewer Cxqe -> Weakness3)**.
>
> ---
>
> > **Reference**
>
> [1] Adams, Lisa, et al. "Longhealth: A question answering benchmark with long clinical documents." Journal of Healthcare Informatics Research 9.3 (2025): 280-296.
>
>
> ---
>
> We appreciate your review once again and hope our responses adequately address your concerns.
>
> If any further questions remain, please do not hesitate to raise them.

---

> > ### Author Rebuttal · Reviewer_Cxqe · 2026-04-02
> >
> > My concerns are addressed.

---

> > > ### Author Response · Authors · 2026-04-03
> > >
> > > Thank you for your reply and for supporting our work! We are confident that integrating your valuable insights will help us improve the manuscript!

---

### Official Review · Reviewer_ocsQ · 2026-03-13

**Soundness:** 3
**Presentation:** 2
**Significance:** 2
**Originality:** 3
**Overall Recommendation:** 5
**Confidence:** 4

**Summary:**

This paper proposes an input-dependent strategy to determine which attention heads will use dense and which will use sparse attention. The idea is to train a router that via gumbel-softmax + STE to predict, based on the input, which heads in attention should be dense or sparse. The authors further add additional terms to the loss function to avoid degenerate scenarios such as fully dense or fully sparse cases. The key empirical claim is that downstream tasks can be grouped into sparsity-robust and sparsity-sensitive categories, and that the model can benefit from adapting its sparsity accordingly. Experiments on LongBench-E, RULER, and LongBench-v2 show competitive results, especially in long-context tasks.

**Compliance With Llm Reviewing Policy:**

Affirmed.

**Final Justification:**

The rebuttal has addressed my concerns. I believe the paper offers substantial novelty along with empirical rigor and good practical implementations.

**Key Questions For Authors:**

- In Section 2.2, could you clarify the preliminary experiment in more detail? In particular: what exact sparse pattern is used when replacing retrieval heads with sparse heads? what task/data setup is used?

- The router is trained with task-dependent sparsity bounds. How should we choose these bounds in a new task where the optimal sparsity is unknown? How sensitive are results to this choice?

- On RULER, your method still remains below the dense backbone at several context lengths. What do you see as the main remaining bottleneck preventing the method from closing the gap to dense attention?

- Do you have results on RULER separated by each subtask? In which tasks Elastic Attention perform better?

- The fused kernel is an interesting contribution. Could you provide more detail on its implementation?

**Limitations:**

See weakness.

**Strengths And Weaknesses:**

Pros:
- Adapting attention sparsity at inference time instead of fixing a single sparse/full ratio for all tasks is a very interesting and timely problem

- The distinction between sparsity-robust and sparsity-sensitive tasks is interesting and useful as a high-level perspective. I really enjoyed the preliminary study in Section 2.2 and I think it's one of the strongest parts of the paper.

- The method is conceptually simple: a lightweight router assigns each head to FA or SA, which makes the approach easy to understand and to apply.

- The evaluation is fairly comprehensive, covering LongBench-E, RULER, and LongBench-v2 across multiple model backbones.

- It is interesting that the router analysis shows some heads becoming consistently routed to sparse mode, while others remain in FA. Though it's likely task-specific, it is a nice finding.

- The paper is honest about at least some weaknesses of the method, e.g. noting that it underperforms baselines on some sparsity-robust tasks such as summarization and code.

Cons:

- The distinction between retrieval heads and sparse heads is not fully convincing to me. Section 2.1 essentially identifies retrieval heads with full attention, but it is not clear why a retrieval head could not also be implemented with a sparse mechanism. That framing feels too rigid, especially in light of recent works such as AdaSplash [1] and InfLLMv2 [2].

- Although I really liked section 2.2, it is a bit hard to interpret the results because the key details are deferred to Appendix C.

- The sparse pattern itself seems limited in flexibility in the main setup: for SSA/XA-based heads, the method still inherits the limitations of the chosen sparse pattern

- The training objective depends on task-specific target sparsity bounds $t$, but the paper does not really explain how to set these bounds in practice. Since the optimal sparsity is unknown, I presume that choosing these bounds can require substantial tuning.

- The paper freezes the backbone and trains only the router. This is computationally attractive, but it also limits what is being shown: it does not demonstrate whether elastic sparsity remains effective under continued pretraining or smaller from-scratch training, which would better test the full potential of the idea.

- On LongBench-E, the results are competitive but not especially decisive. That is, there's no clear winner in Table 1.

- On RULER, the sparse method is still generally below the dense backbone, especially at several context lengths, so the paper does not fully close the quality gap to dense attention even though it often improves over sparse baselines. However, the results are much more promising here.

- The extrapolation aspect in RULER could be presented more clearly. For example, by adding marks on >64K to denote extrapolation regime.

- The kernel discussion is still somewhat incomplete from a systems perspective. The paper claims a fused kernel for jointly processing routed heads, but some important implementation questions are unclear, like how the kernel handles non-contiguous sparse heads and memory coalescing.

[1] Gonçalves, Nuno, Marcos Treviso, and André FT Martins. "Adasplash: Adaptive sparse flash attention." ICML 2025
[2] Zhao et al. "InfLLM-V2: Dense-Sparse Switchable Attention for Seamless Short-to-Long Adaptation". ICLR 2026

---

> ### Author Rebuttal · Authors · 2026-03-29
>
> Thanks for your review. Due to the rebuttal word limit, we provide the anonymous URL (https://anonymous.4open.science/r/Elastic-Attention-D370/rebuttal/rebuttal.md) containing complete figures and tables.
>
> ---
>
> > **W1**: The dichotomy between "retrieval heads" (full attention) and "sparse heads" is too rigid.
>
> **R**: Thanks for raising this point. There is a misunderstanding. (1) In Section 5.4, we explore deploying retrieval heads with Sparse Attention; (2) As noted in Lines 98-99, we deliberately employ the qualifiers "in practice" and "typically" to indicate this is a tendency; (3) Consistent with established works such as DuoAttention. We will still polish this writing in the final version!
>
> ---
>
> > **W2 & W8**: Key details deferred to Appendix C hinder result interpretation. The RULER extrapolation regime (>64K) needs clearer visual demarcation.
>
> **R**: Thanks for your appreciation of Section 2.2. We commit to incorporating key results from Table 5 (Appendix C) into a dedicated table within Section 2.2 in the final version (which permits an additional page). Furthermore, we will use distinct markers to indicate extrapolation for RULER results exceeding 64k.
>
> ---
>
> > **W4 & Q2**：The training objective depends on task-specific sparsity bounds, yet the paper lacks practical guidance for setting them. Without knowing the optimal sparsity, bound selection likely requires extensive tuning—raising concerns about applicability to new tasks.
>
> **R**: Thank you for this excellent question. We have addressed this question in our response to **Reviewer 6by5 (W2 & Q2)**. Please refer to that reply for detailed explanations!
>
> ---
>
> > **W5**：While computationally efficient, freezing the backbone limits the study—it fails to validate whether elastic sparsity works under continued pretraining or from-scratch training.
>
> **R**: We appreciate this valuable suggestion. However, training from scratch would contradict our core motivation: pre-trained models already possess inherent retrieval head capabilities. We instead explored continued pretraining using both LoRA and full fine-tuning. **We put the full table in the anonymous repository (rebuttal.md -> Reviewer ocsQ -> weakness5)**. We observe that utilizing more trainable parameters significantly improves performance on real-world tasks (LongBench), while synthetic long-context retrieval capabilities remain consistent with the results reported in the paper.
>
> ---
>
> > **W7 & Q4**: Requirement of results on RULER separated by each subtask, and in which tasks Elastic Attention performs better.
>
> **R**: Elastic Attention significantly narrows the quality gap compared to prior sparse methods. As shown in the **full comparison table in our anonymous repository (rebuttal.md -> Reviewer ocsQ -> weakness7)**: (1) on Llama3.1-8B-Ins, our FA-XA achieves only a -1.64 average gap (compared to -24.37 for InfLLM-V2 and -20.54 for DuoAttention); (2) It specifically excels on single-needle retrieval tasks (e.g., niah_single_2, niah_single_1), where it matches or even exceeds dense performance.
>
> ---
>
> > **W9 & Q5**：The kernel discussion is somewhat incomplete from a systems perspective.
>
> **R**: Thanks for raising this question. Our fused kernel enables joint parallel processing of FA and SA routed heads in one forward pass, eliminating multi-kernel launch and tensor split/concat overhead. Each attention head maps to an independent CUDA Thread Block, with block-level branching eliminating warp divergence, no memory rearrangement, and no performance loss from non-contiguous sparse heads. Q/K/V tensors maintain a contiguous, aligned HBM layout, and our block-based contiguous I/O fully complies with GPU memory coalescing rules.
> **Full pseudocode and CUDA implementation are available at the anonymous link (rebuttal.md -> Reviewer ocsQ -> weakness9).**
>
> ---
>
> > **Q1**: Clarify the preliminary experiment in Section 2.2 with more detail.
>
> **R**: You might have missed the preliminary study details shown in Appendix C. We adopt the SSA pattern (Line 753), and evaluate model on LongBench (Line 157-158). For sparse head detection, we show details in Appendix C.1 and C.2.
>
> ---
>
> > **Q3**: Discussion about main remaining bottleneck preventing sparse attention from closing the gap to dense attention.
>
> **R**: We view this as an inherent trade-off consistent with the No Free Lunch theorem: reducing computational costs (FLOPs) naturally incurs some representational capacity cost. However, we argue this gap is practically mitigated by two factors: (1) in ultra-long contexts, sparse attention alleviates softmax dilution issues and effectively filters contextual noise in dense attention; (2) our hybrid architecture employs full attention as a safety module, which inherently bounds the performance gap while preserving the efficiency gains.
>
> ---
>
> We appreciate your review once again and hope our responses adequately address your concerns.
>
> If any further questions remain, please do not hesitate to raise them.

---

> > ### Author Rebuttal · Reviewer_ocsQ · 2026-04-03
> >
> > Thanks for all clarifications. I am still learning towards a weak accept.
> >
> > Can you provide the actual kernel implementation details here?
> >
> > NOTE: I did not read the content in the mentioned URL as per ICML rules:
> >
> > > [...] the response should not contain non-anonymized URLs, URLs for personal websites, or “shortened” URLs (e.g., as provided via tinyurl, which could log a reviewer’s IP). Reviewers are not expected to follow external URLs in the response.

---

> > > ### Author Response · Authors · 2026-04-03
> > >
> > > Thank you for acknowledging our clarifications.
> > >
> > > Regarding the ICML 2026 rules (https://icml.cc/Conferences/2026/AuthorInstructions), we respectfully note that the restriction applies specifically to **non-anonymized URLs**. The link we provided is fully anonymized, and thus complies with the policy.
> > >
> > > As the Peer Review FAQ (https://icml.cc/Conferences/2026/PeerReviewFAQ) states,
> > >
> > > > links should be used "primarily for figures (including tables) and captions that describe the figure," serving only to provide missing details rather than present new work.
> > >
> > > We assure you that we have not modified any paper content; we only provide additional figures/tables as requested.
> > >
> > > ---
> > >
> > > Due to Markdown rendering constraints, pseudo-code cannot be displayed. The algorithm workflow is therefore presented in plain text below:
> > >
> > > ps. We strongly recommend referring to the anonymous link submitted with the original paper (not added during rebuttal) for the complete context.
> > >
> > > ---
> > >
> > > **1. Parallelization & Dispatching**: The computation is parallelized across batches, attention heads, and query blocks. The kernel reads the head mask type (DENSE, SPARSE, or STREAMING) to dispatch the corresponding processing routine. By operating at the thread-block level, this heterogeneous dispatch guarantees zero intra-warp divergence.
> > >
> > > **2. Memory Initialization**: For each target query block, data is loaded from HBM into SRAM. To support the FlashAttention-style online softmax, running statistics (local maximum and sum) and the output tensor are initialized directly in fast hardware registers.
> > >
> > > **3. Pointer Leaping Mechanism**: To avoid redundant computation, the kernel evaluates the block mask to construct an ordered set of valid Key/Value (K/V) blocks. It computes exact pointer offsets to "leap" over masked regions, iterating exclusively over valid indices.
> > >
> > > **4. Coalesced Fetch & Tensor Core Computation**: Within the sparse inner loop, valid K/V blocks are fetched from HBM to SRAM using coalesced, 128-bit vectorized memory copies. The block-wise attention scores are then computed efficiently leveraging Tensor Core MMA instructions.
> > >
> > > **5. Online Softmax & Accumulation**: Following the computation of local attention scores, the kernel dynamically updates the running maximum, rescales the historical accumulators, computes the local exponential sum, and aggregates the weighted values into the output registers.
> > >
> > > **6.Finalization**: Upon completing the sparse iteration, the fully accumulated output block is normalized using the total running sum and written back to HBM.
> > >
> > > ---
> > >
> > > In summary, our fused kernel is specifically optimized to avoid common pitfalls like warp divergence and uncoalesced memory I/O. We will add the following description in the revised manuscript:
> > >
> > > **1. Handling Non-contiguous Sparse Heads (Zero Warp Divergence)**: Heterogeneous head processing is dispatched at the Thread-Block level. Because each CUDA thread block corresponds to a specific head, evaluating the head mask type (Dense vs. Sparse) at the block entry ensures that all threads within a block (and thus within a warp) execute the exact same instruction path. This completely eliminates intra-warp divergence while allowing dense and sparse heads to be processed concurrently across the GPU grid.
> > >
> > > **2. Memory Coalescing via Pointer Leaping**: For sparse heads, the kernel does not read masked K/V blocks and then discard them using conditional branches (which would waste memory bandwidth). Instead, we designed an iterator (fwdIterator in our code) that pre-computes a leap offset to the next valid block. The kernel directly advances the global memory pointers by this offset. When a valid block is fetched, the loads are performed using cute::TiledCopy (from NVIDIA's CuTe library), ensuring 128-bit vectorized, perfectly coalesced memory accesses for the entire block. Thus, jumping over sparse regions preserves high memory bandwidth utilization.
> > >
> > > ---
> > >
> > > Once again, thank you for your thorough and detailed questions. Please feel free to reach out if you have any further inquiries.

---

### Decision · Program_Chairs · 2026-04-30

**Decision:**

Accept (regular)

**Comment:**

The paper introduces "Elastic Attention," an adaptive hybrid attention mechanism designed to accelerate long-context LLM inference. It utilizes a lightweight, input-conditioned router to dynamically assign individual attention heads to either Full or Sparse Attention at test time.

Reviewers unanimously recommended the paper for acceptance. They recognize the importance of the problem, the technical contributions and results. The rebuttal has resolved concerns.

Overall this is a strong work with high practical values.